# FROM INPAINTING TO EDITING: A SELF-BOOTSTRAPPING PARADIGM FOR CONTEXT-RICH VISUAL DUBBING

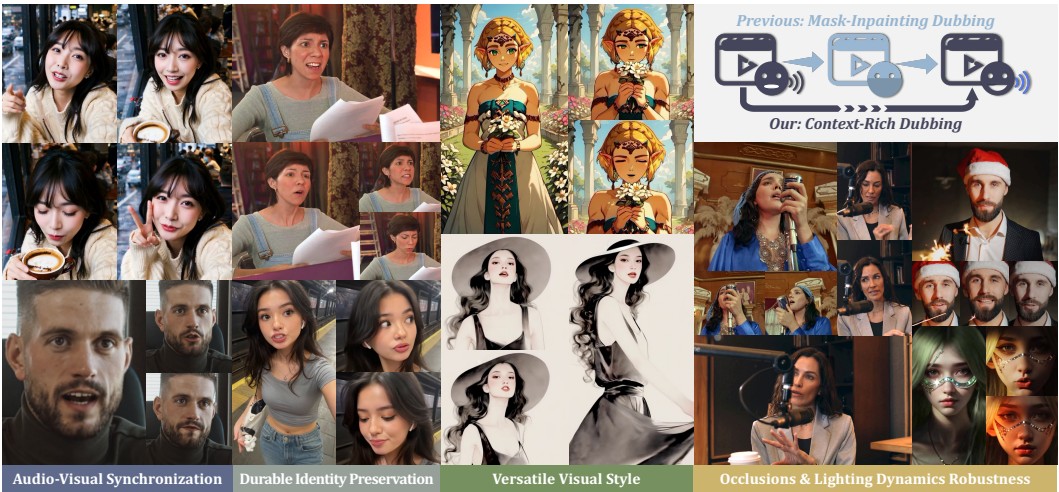

Figure 1: Moving beyond mask-inpainting, **X-Dub** redefines visual dubbing as context-rich, full-reference video-to-video editing, which yields precise lip synchronization and faithful identity preservation, even in challenging scenarios with occlusions and dynamic lighting.

## ABSTRACT

Audio-driven visual dubbing aims to synchronize a video's lip movements with new speech, but is fundamentally challenged by the lack of ideal training data: paired videos where only a subject's lip movements differ while all other visual conditions are identical. Existing methods circumvent this with a mask-based inpainting paradigm, where an incomplete visual conditioning (i.e., masked video frames and misaligned appearance references) forces models to simultaneously hallucinate missing content and sync lips, leading to visual artifacts, identity drift, and poor synchronization. In this work, we propose a novel self-bootstrapping framework that reframes visual dubbing from an ill-posed inpainting task into a well-conditioned video-to-video editing problem. Our approach employs a Diffusion Transformer (DiT), first as a data *generator*, to synthesize ideal training data: a lip-altered companion video for each real sample, forming visually aligned video pairs. A DiT-based audio-driven *editor* is then trained on these pairs end-to-end, leveraging the complete and aligned input video frames to focus solely on precise, audio-driven lip modifications. This complete, frame-aligned input conditioning forms a rich *"visual context"* for the editor, providing it with complete identity cues, scene interactions (e.g., lighting and occlusions), and continuous spatiotemporal dynamics. Leveraging this rich context fundamentally enables our method to achieve highly accurate lip sync, faithful identity preservation, and exceptional robustness against challenging in-the-wild scenarios. We further introduce a timestep-adaptive multi-phase learning strategy as a necessary component to disentangle conflicting editing objectives across diffusion timesteps, thereby facilitating stable training and yielding enhanced lip synchronization and visual

fidelity. Additionally, we propose ContextDubBench, a comprehensive benchmark dataset for robust evaluation in diverse and challenging practical application scenarios. Our visualizations are available at the anonymous page x-dub-lab.github.io, and code will be released to benefit the community.

# 1 INTRODUCTION

Audio-driven visual dubbing aims to synchronize an existing video's lip movements with new speech (KR et al., 2019). Unlike audio-driven animation (Tian et al., 2024; Cui et al., 2024a), which generates entire videos from scratch, dubbing requires modifying only the speech-relevant regions while strictly preserving identity and other visual cues from the original video. This uniqueness underpins dubbing's broad applications, from personalized avatars (Thies et al., 2020) to multilingual film translation (Prajwal et al., 2020). The recent rise of Diffusion Transformers (DiTs) (Peebles & Xie, 2023) has demonstrated their remarkable generation capability in text-to-video (T2V) and image-to-video (I2V) tasks, making them natural candidates for high-fidelity visual dubbing. Yet dubbing presents a distinctive challenge. As a video-to-video task, it ideally requires paired training data where lip movements differ while identity, pose, and environment remain unchanged; obtaining such paired data is physically impossible in the real world.

This fundamental data constraint has historically forced existing approaches into a **mask-inpainting paradigm** (Prajwal et al., 2020): they mask the lower half of the face in a video and train a model to inpaint it conditioned on the corresponding audio and sparse reference frames. While enabling self-supervised reconstruction without paired data, this design inevitably degrades visual conditioning by stripping visual information from the lower facial region and relying on often misaligned, sparse references. Consequently, the inpainting model must not only reconstruct synchronized lip motion, but simultaneously hallucinate missing content (e.g., facial occlusions absent in references) and extract identity appearance from pose-misaligned reference frames. This divided attention compromises precise lip synchronization, further exacerbated by mask-boundary lip-shape leakage and weak audio conditioning (Wang et al., 2023; Chen et al., 2025), while also inducing identity drift and visual artifacts (Zhong et al., 2023; Peng et al., 2025). Ultimately, these failures reveal a fundamental disconnect: the training paradigm renders the task *ill-posed* by restricting the *"visual context"*—the visual conditioning information available to the model—to a fragmented state (i.e., masked frames and misaligned references), preventing it from leveraging the complete video context (i.e., the full input video) that is actually available during inference. Therefore, we argue that achieving high-quality visual dubbing does not hinge on improving existing inpainting architectures, but on fundamentally breaking away from this mask-inpainting paradigm.

We therefore propose a new perspective that directly confronts this data bottleneck **to restore the task to its intrinsic editing nature**. Instead of circumventing the problem with the mask-inpainting paradigm, we present **X-Dub**, a novel self-bootstrapping framework that leverages the generative priors of a pre-trained DiT model to synthesize ideal paired training data for itself. Specifically, a DiT-based *generator* first synthesizes a lip-altered companion video for each real sample, creating visually aligned pairs that share identical identity, pose, and scene but differ in lip motion. A subsequent DiT-based *editor* then learns dubbing directly from these video pairs, taking the full companion video as input to perform mask-free editing. In this way, we transform dubbing from an *ill-posed inpainting task* into a *well-conditioned editing problem* guided by complete, frame-aligned visual context, fundamentally aligning the solution with the definition of visual dubbing.

Concretely, the *generator* is instantiated akin to a conventional mask-inpainting model trained on large-scale audiovisual data via self-reconstruction. Once trained, it operates offline to inpaint the masked source video conditioned on alternative audio, producing a lip-altered companion to construct a synthetic-real training pair. Crucially, within our framework, the *generator* functions not as the final dubbing solver but as a contextual condition synthesizer. This role shift allows us to make a principled trade-off: we optimize primarily for identity preservation and spatiotemporal robustness, minimizing artifacts even at the cost of perfect lip-sync accuracy. This yields a synthetic companion that serves as a rich *"visual context"*, providing all visual information aligned with the target real video—including rich identity cues, complete scene interactions (e.g., lighting and occlusions), and continuous spatiotemporal dynamics—except for the target lip motion. Consequently, learning from these aligned pairs, the *editor* faces a significantly simplified task: it can dedicate its full capacity to

precise speech-driven lip editing while seamlessly inheriting all other visual factors from this complete context, ultimately achieving superior lip-sync accuracy, identity preservation, and robustness to visual variations.

Reframing visual dubbing as targeted editing, however, introduces unique conflicting goals: the model must simultaneously inherit global structure, edit local lips, and preserve fine-grained texture. A monolithic learning approach often struggles to balance these objectives. We therefore introduce a **timestep-adaptive multi-phase learning** strategy with LoRA experts as a necessary component for the *editor*. We leverage the inherent tendency of diffusion models to capture distinct information levels at different timesteps (Zhang et al., 2025; Wang et al., 2025; Liang et al., 2025; Peng et al., 2025). By aligning early, mid, and late diffusion stages with global structure, lip shape, and texture refinement, respectively, we disentangle these conflicting objectives, allowing the model to learn complementary tasks at their most effective phases, thereby facilitating stable training and yielding enhanced lip-sync accuracy and visual quality.

Finally, we note that existing visual dubbing benchmarks (Afouras et al., 2018; Zhang et al., 2021) are confined to controlled settings with limited motion and diversity, insufficient for evaluating robustness in realistic scenarios. We therefore introduce **ContextDubBench**, a benchmark built from both real-world footage and advanced generative content, encompassing varied motions, environments, styles, and subjects to enable comprehensive evaluation under complex dubbing conditions.

In summary, our contributions are: 1) We propose a **self-bootstrapping visual dubbing framework** that leverages pre-trained DiT models both as a data *generator* to synthesize aligned training video pairs and as an audio-driven *editor* trained on them for the final dubbing task, transforming the task from an ill-posed inpainting problem into a well-conditioned editing one guided by complete visual context. 2) We introduce a **timestep-adaptive multi-phase learning** strategy as a necessary component to disentangle conflicting editing objectives across diffusion timesteps, facilitating stable contextual learning and yielding enhanced lip sync and visual fidelity. 3) We construct and release **ContextDubBench**, a comprehensive benchmark specifically designed to evaluate visual dubbing models in challenging real-world and generative scenarios. 4) Extensive experiments demonstrate that our method achieves remarkable improvements across all metrics, significantly outperforming existing approaches with more accurate lip sync, superior identity preservation, and exceptional robustness to spatiotemporal variations.

## 2 RELATED WORK

**Visual dubbing.** Early visual dubbing methods leverage GANs (Goodfellow et al., 2014) for mask-based inpainting. LipGAN (KR et al., 2019) pioneers this direction with reference-guided synthesis, while Wav2Lip (Prajwal et al., 2020) improves lip sync through SyncNet (Chung & Zisserman, 2016). Subsequent works extend this paradigm: VideoReTalking (Cheng et al., 2022) introduces canonical references to mitigate expression bias, DINet (Zhang et al., 2023) enables high-resolution synthesis via deformation inpainting, and TalkLip (Wang et al., 2023) enhances lip intelligibility using AV-HuBERT (Shi et al., 2022). IP-LAP (Zhong et al., 2023) and StyleSync (Guan et al., 2023) further strengthen identity preservation through landmark- and style-aware optimization.

Recent diffusion-based approaches also exhibit advanced performance. DiffTalk (Shen et al., 2023) and Diff2Lip (Mukhopadhyay et al., 2024) demonstrate the feasibility of diffusion, while MuseTalk (Zhang et al., 2024) achieves real-time synthesis by combining latent diffusion with adversarial training. LatentSync (Li et al., 2024) adapts pre-trained diffusion models with temporal supervision to improve stability. Nevertheless, these methods largely follow a self-reconstruction paradigm based on masked frames and sparse references, which limits contextual richness. By contrast, our approach introduces a *contextual conditioning paradigm*, where paired videos provide informative context, allowing the model to focus on accurate lip editing with stronger stability.

**Audio-driven portrait animation.** Another related line of work is audio-driven portrait animation, which generates talking videos from still images or text prompts. Recent DiT-based models achieve expressive talking-head (Tian et al., 2024; Cui et al., 2024a), half-body (Cui et al., 2024b; Meng et al., 2025), and full-body results (Wang et al., 2025; Lin et al., 2025). These works demonstrate the power of DiTs for human-centric generation in I2V or T2V paradigms. Visual dubbing instead is a

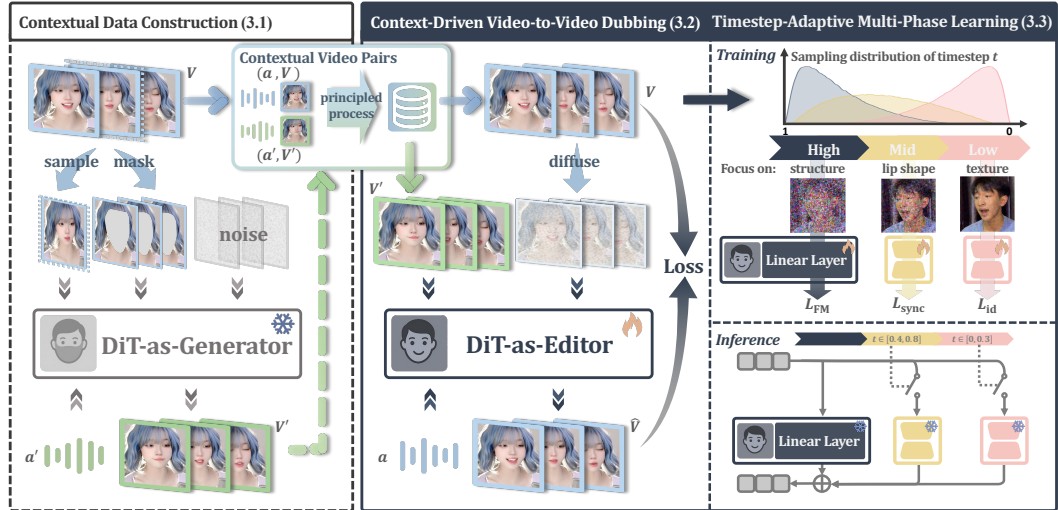

Figure 2: **Overview of X-Dub, our self-bootstrapping dubbing framework.** At its core, our paradigm employs a DiT *generator* to create a lip-altered counterpart for each video, forming a context-rich pair with the original (left). A DiT *editor* then learns mask-free, video-to-video dubbing directly from these ideal pairs, leveraging the complete visual context to ensure accurate lip sync and identity preservation (middle). This contextual learning is further refined by our timestep-adaptive multi-phase learning (right), which aligns different diffusion stages with learning distinct information: global structure, lip movements, and texture details, respectively.

stricter video-to-video editing task: it requires precise speech-driven modifications while preserving other visual cues, enabling seamless integration into recorded videos.

## 3 OUR APPROACH

As illustrated in Fig. 2, we establish a self-bootstrapping dubbing framework where a DiT model both generates visually aligned video pairs with varied lip motion and learns dubbing from them, thereby reframing dubbing from an ill-posed inpainting problem into a well-conditioned video-to-video editing task. We first present the DiT-based *generator*, trained with a mask-inpainting self-reconstruction objective to synthesize lip-varied companion videos that serve purely as contextual inputs (Sec. 3.1). To ensure these synthetic companions serve as reliable visual conditioning, we introduce principled construction strategies that prioritize identity preservation and robustness over secondary lip accuracy and generalization, further employing rigorous quality filtering and augmentations to minimize artifacts and maximize visual alignment (Sec. 3.1.2). On top of such curated video pairs, the DiT-based *editor* learns mask-free dubbing as rich-context-driven editing, achieving accurate lip sync, faithful identity retention, and resilience to pose and occlusion variations (Sec. 3.2). Finally, we propose a timestep-adaptive multi-phase learning scheme (Sec. 3.3) that aligns diffusion stages with complementary objectives—structure, lips, and textures—to facilitate stable training convergence within this editing paradigm and further enhance dubbing quality.

**DiT backbone.** Our DiT backbone follows the latent diffusion paradigm with a 3D VAE for video compression and a DiT for token sequence modeling (Peebles & Xie, 2023). Each DiT block combines 2D spatial and 3D spatio-temporal self-attention with cross-attention for external conditions.

### 3.1 GENERATOR: CONTEXTUAL CONDITION CONSTRUCTOR

#### 3.1.1 NAÏVE MASK DUBBING

We implement the DiT-based *generator* under a mask-based self-reconstruction scheme following prior dubbing methods. Given a target video $V_{tgt}$ with audio $a_{tgt}$, we apply a facial mask $M$ and reconstruct masked regions $\hat{V}_{tgt}$ conditioned on $a_{tgt}$ and a reference frame $I_{ref}$.

Although this setup yields imperfect dubbing outputs, the *generator* is not designed to solve dubbing directly, but solely to synthesize companion videos as contextual inputs for the *editor* in our framework. By altering lip motion within otherwise consistent frames, the *generator* transforms sparse inpainting contexts into aligned video pairs far stronger than static reference frames.

**Conditioning mechanisms.** As shown in Fig. 3, masked and target frames are encoded by a VAE into $z_{\text{mask}}, z_{\text{tgt}} \in \mathbb{R}^{b \times f \times c \times h \times w}$, and the reference frame into $z_{\text{ref}} \in \mathbb{R}^{b \times c \times h \times w}$. We concatenate $z_{\text{mask}}$ with noised $z_{\text{tgt}}$ channel-wise, and zero-pad $z_{\text{ref}}$ for channel alignment. Concatenating across frames yields the unified DiT input $z_{\text{in}} = \left[ [\mathbf{0}, z_{\text{ref}}]_{\text{ch}}, [z_{\text{mask}}, z_{\text{tgt}}]_{\text{ch}} \right]_{\text{fr}}$, which enables interaction between video and reference tokens via 3D self-attention. Whisper (Radford et al., 2023) features are injected through cross-attention as audio condition. To extend generation to long videos, we use motion frames (Tian et al., 2024): each segment is conditioned on the last frames of the previous one. During training, the first $m = 2$ frames of $z_{\text{tgt}}$ remain unnoised as motion guidance. Conditional dropout (50%) handles the absence of prior frames in initial segments.

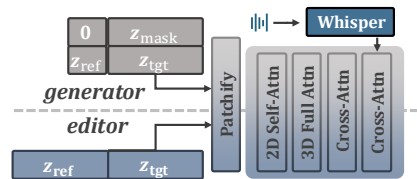

Figure 3: **Conditioning mechanisms for our DiT-based framework**. Reference conditions (full contextual video frames for *editor*; a single reference frame for *generator*) and the target video are concatenated into a unified sequence for 3D self-attention. Audio is injected via cross-attention.

**Training objective.** We adopt a flow-matching loss $\mathcal{L}_{\text{FM}}$ (details in Appendix. B.1), weighted by face and lip masks $M, M_{\text{lip}}$ from DWPose (Yang et al., 2023) via element-wise multiplication ($\odot$):

$$\mathcal{L}_{\text{wFM}} = (1 + \lambda M + \lambda_{\text{lip}} M_{\text{lip}}) \odot \mathcal{L}_{\text{FM}}. \tag{1}$$

Trained in this manner, the *generator* produces a synthetic companion video $V'$ for each real clip $V$ by replacing its original audio $a$ with an alternative $a'$, yielding frame-aligned but lip-varied paired videos $(V', V)$. Here, $V'$ serves solely as the conditional input for the *editor*.

### 3.1.2 PRINCIPLED PAIR CONSTRUCTION STRATEGIES

Plain mask-based dubbing inevitably yields imperfect results. We therefore design explicit trade-off strategies within our *generator*'s data construction process to ensure that the synthetic companion video, while not flawless, serves as a reliable contextual input, turning the *generator* from a naïve dubber to a reliable contextual condition provider.

To this end, we establish three guiding principles for our data construction process: **1) In-domain quality over generalization.** The *generator* should focus on fidelity within the training distribution rather than broad generalization. **2) Visual consistency under variation.** Companion videos must maintain identity and remain robust to pose, occlusion, and illumination changes. **3) Lip variation over accuracy.** Lip shapes in $V'$ should differ from $V$ to avoid leakage, while tolerating moderate lip-sync inaccuracies.

Accordingly, we implement several strategies specifically for the *generator* during the paired data construction phase. First, we leverage short-term visual stationarity by having the *generator* process videos in brief, 25-frame segments where pose and scene remain relatively stable. During this data-creation inference, intra-segment reference frames are selected to provide roughly aligned visual information for each shorter clip. Motion frames then connect these segments into complete 77-frame videos for the subsequent *editor* training, which exhibit minimal identity and color drift as a result of this process. While lip accuracy may degrade across segment boundaries, this trade-off favors visual consistency as intended. We also sample alternative audio $a'$ from the same speaker as $V$ to reduce cross-identity conflicts and apply **extended training** beyond nominal convergence to improve identity preservation and lip sync.

To enhance robustness to diverse variations, we incorporate complementary techniques. We handle **occlusions** by annotating and excluding facial occluders from inpainting areas, enabling the generator to be more robust to occlusion scenarios. For **lighting augmentation**, we apply identical relighting to both $V$ and $V'$ in uniformly-lit videos to construct pairs with consistent lighting dynamics. We also perform **quality filtering** using landmark distance, identity similarity, and overall visual

quality scores to ensure sufficient lip divergence, identity preservation, and high visual fidelity. Additionally, we supplement with **3D-rendered data** to obtain perfectly aligned pairs. Implementation details are in Appendix C.

Together, these data construction strategies ensure the *generator* produces lip-varied video pairs that, though not perfect, consistently provide strong and reliable conditions for the *editor*.

## 3.2 EDITOR: CONTEXT-DRIVEN VIDEO-TO-VIDEO DUBBER

Given curated pairs $(V', V)$, we train a DiT-based *editor* for mask-free dubbing. Unlike the *generator*, the *editor* tackles dubbing directly: given audio $a$ and the companion video $V'$, it learns to produce $V$ as the target, thereby transforming dubbing from a sparse inpainting problem into context-driven editing. In practice, the *editor* surpasses the *generator* across lip accuracy, identity preservation, and robustness, benefiting from the rich contextual input provided by the paired videos.

**Contextual conditioning mechanisms.** As shown in Fig. 3, the paired reference and target videos are encoded as latents $z_{\text{ref}}, z_{\text{tgt}} \in \mathbb{R}^{b \times f \times c \times h \times w}$. The diffused $z_{\text{tgt}}$ is then concatenated with clean $z_{\text{ref}}$ across frames, forming $z_{\text{in}} \in \mathbb{R}^{b \times 2f \times c \times h \times w}$. Patchifying this sequence enables contextual interaction via 3D self-attention, minimally altering the DiT backbone while fully exploiting its contextual modeling capacity. Audio features and motion frames are integrated identically to Sec. 3.1.

## 3.3 TIMESTEP-ADAPTIVE MULTI-PHASE LEARNING WITH LoRA EXPERTS

While lip-varied video pairs significantly simplify dubbing, training the *editor* must still balance inheriting global structure, editing lip motion, and preserving fine-grained identity details. Diffusion models exhibit stage-wise specialization across timesteps (Zhang et al., 2025; Wang et al., 2025), motivating us to introduce a timestep-adaptive multi-phase scheme, where different noise regions target complementary objectives.

**Phase partitioning.** Following Esser et al. (2024), we shift the timestep sampling distribution to concentrate on different noise levels for each training phase:

$$t_{\text{shift}} = \frac{\alpha t_{\text{orig}}}{1 + (\alpha - 1)t_{\text{orig}}}, \tag{2}$$

where $t_{\text{orig}}$ is logit-normal and $\alpha$ sets the shift strength. This yields: 1) high-noise steps for global structure and motion, including background layout, head pose, and coarse identity; 2) mid-noise steps for lip movements; 3) low-noise steps for texture refinement concerning identity details.

**High-noise full training.** We first train the *editor* under the high-noise distribution with full-parameter optimization. This setting not only facilitates convergence and improves generation quality (Esser et al., 2024), but also encourages the model to learn global structures effectively, thus seamlessly transferring background, head pose, overall identity, and other spatiotemporal dynamics from reference contexts while achieving preliminary lip sync. The training objective is the same mask-weighted flow-matching loss $\mathcal{L}_{\text{wFM}}$ as in Eq. 1.

**Mid- and low-noise tuning with LoRA experts.** We then attach lightweight LoRA modules for mid- and low-noise phases. Since pixel-level constraints are needed, we design a single-step denoising strategy to avoid computational overhead during training:

$$\hat{x}_0 = \mathcal{D}(z_0 + (v - \hat{v}) \cdot \min\{t, t_{\text{thres}}\}), \tag{3}$$

where $t_{\text{thres}}$ ensures stable denoising at high noise levels (see Appendix D for detailed derivation).

The *lip expert* operates at mid-noise, supervised by an additional lip-sync loss $\mathcal{L}_{\text{sync}}$ using Sync-Net (Chung & Zisserman, 2016) for audio-visual alignment. The *texture expert* works at low-noise with identity loss $\mathcal{L}_{\text{id}}$ computed against references using ArcFace (Deng et al., 2019) and CLIP (Radford et al., 2021) features. To avoid hurting sync, we randomly disable audio cross-attention (probability 0.5) during texture tuning, computing texture supervision only under silent conditions.

During inference, we activate each LoRA within its optimal timestep range: texture expert for $t \in [0, 0.3]$ and lip expert for $t \in [0.4, 0.8]$, ensuring each contributes where most effective. Details on the selection of timestep ranges can be found in Appendix D.3.

Table 1: **Quantitative results on HDTF.** Top three are highlighted as first , second , and third .

| | HDTF Dataset | | | | | | | | |
|---|---|---|---|---|---|---|---|---|---|
| | Visual Quality | | | | Lip Sync | | Identity | | |
| Method | PSNR ↑ | SSIM ↑ | FID ↓ | FVD ↓ | Sync-C ↑ | LMD ↓ | LPIPS ↓ | CSIM ↑ | CLIPS ↑ |
| Wav2Lip | 27.412 | 0.851 | 15.475 | 530.905 | 7.663 | 0.896 | 0.078 | 0.807 | 0.842 |
| VideoReTalking | 25.189 | 0.844 | 11.303 | 327.886 | 7.482 | 1.170 | 0.056 | 0.745 | 0.808 |
| TalkLip | 27.024 | 0.850 | 17.315 | 564.307 | 5.887 | 0.858 | 0.060 | 0.804 | 0.855 |
| IP-LAP | 28.571 | 0.860 | 9.026 | 352.403 | 5.199 | 0.934 | 0.041 | 0.840 | 0.899 |
| Diff2Lip | 28.716 | 0.860 | 12.251 | 348.290 | 7.897 | 0.911 | 0.036 | 0.790 | 0.876 |
| MuseTalk | 29.542 | 0.866 | 8.123 | 258.236 | 6.409 | 0.741 | 0.029 | 0.824 | 0.884 |
| LatentSync | 31.325 | 0.903 | 8.042 | 235.524 | 8.163 | 0.821 | 0.024 | 0.847 | 0.902 |
| Ours-*generator** | 34.253 | 0.914 | 7.873 | 172.520 | 8.045 | 0.670 | 0.018 | 0.855 | 0.917 |
| **Ours-*editor*** | 34.425 | 0.934 | 7.031 | 176.630 | 8.562 | 0.630 | 0.014 | 0.883 | 0.923 |

Table 2: **Quantitative results on ContextDubBench.** "Ref." is short for reference.

| | ContextDubBench | | | | | | | | |
|---|---|---|---|---|---|---|---|---|---|
| | Visual Quality (Ref.) | | Visual Quality (No Ref.) | | | Lip Sync | Identity | | Generation |
| Method | FID ↓ | FVD ↓ | NIQE ↓ | BRISQUE ↓ | HyperIQA ↑ | Sync-C ↑ | CSIM ↑ | CLIPS ↑ | Sucess Rate ↑ |
| Wav2Lip | 19.330 | 631.589 | 6.908 | 48.397 | 35.667 | 5.087 | 0.738 | 0.805 | 62.95% |
| VideoReTalking | 17.535 | 341.951 | 6.392 | 43.112 | 44.826 | 5.126 | 0.684 | 0.793 | 59.09% |
| TalkLip | 21.262 | 550.658 | 6.284 | 38.990 | 34.311 | 3.213 | 0.739 | 0.724 | 70.45% |
| IP-LAP | 14.891 | 328.728 | 6.576 | 44.879 | 38.059 | 2.292 | 0.797 | 0.809 | 57.73% |
| Diff2Lip | 17.126 | 378.527 | 6.554 | 44.059 | 36.872 | 4.702 | 0.705 | 0.799 | 71.82% |
| MuseTalk | 17.519 | 294.312 | 6.552 | 43.778 | 42.335 | 2.205 | 0.672 | 0.753 | 60.00% |
| LatentSync | 13.602 | 265.057 | 6.113 | 39.154 | 41.654 | 6.282 | 0.801 | 0.812 | 59.77% |
| Ours-*generator** | 10.824 | 224.893 | 5.920 | 36.840 | 48.120 | 6.514 | 0.814 | 0.818 | 66.05% |
| **Ours-*editor*** | 9.351 | 214.298 | 5.782 | 29.870 | 51.960 | 7.282 | 0.850 | 0.839 | 96.36% |

## 4 EXPERIMENTS

**Benchmark.** To evaluate visual dubbing in practical settings, we construct ContextDubBench, a challenging benchmark of 440 video-audio pairs combining real-world and AI-generated content. Videos feature challenging scenarios like profile views, pose shifts, occlusions, and stylized appearances, while audio includes speech and singing across six languages. Unlike existing controlled-environment datasets, it enables evaluation under complex, realistic conditions, as detailed in Sec. I.

**Evaluation metrics.** We evaluate generation quality using PSNR, SSIM, Fréchet Inception Distance (FID) for spatial quality, and Fréchet Video Distance (FVD) for temporal consistency. Lip-sync quality is measured by landmark distance (LMD) and SyncNet confidence (Sync-C). Identity preservation is assessed through cosine similarity of ArcFace embeddings (CSIM), CLIP score (CLIPS) for semantic features, and LPIPS for perceptual similarity.

For the more challenging ContextDubBench, we additionally report no-reference perceptual quality metrics, including Natural Image Quality Evaluator (NIQE), Blind/Referenceless Image Spatial Quality Evaluator (BRISQUE), and HyperIQA (Su et al., 2020). We also report the overall success rate across all 440 video samples, with failed or entirely unsynchronized generations manually excluded. This metric is crucial for practical dubbing scenarios, as traditional methods often fail completely under visual challenges like stylized characters, occlusions, or extreme poses.

### 4.1 QUANTITATIVE EVALUATION

We evaluate our *editor* on both HDTF (Zhang et al., 2021) and ContextDubBench, comparing against state-of-the-art methods including Wav2Lip (Prajwal et al., 2020), VideoReTalking (Cheng et al., 2022), TalkLip (Wang et al., 2023), IP-LAP (Zhong et al., 2023), Diff2Lip (Mukhopadhyay et al., 2024), MuseTalk (Zhang et al., 2024), and LatentSync (Li et al., 2024). We also re-implement a generalizable variant of our *generator*, denoted as *generator**, by removing data-creation-specific constraints and aligning its setup with the *editor*. This allows a fair comparison between traditional inpainting and our context-rich editing dubbing, isolating paradigm gains from backbone capacity.

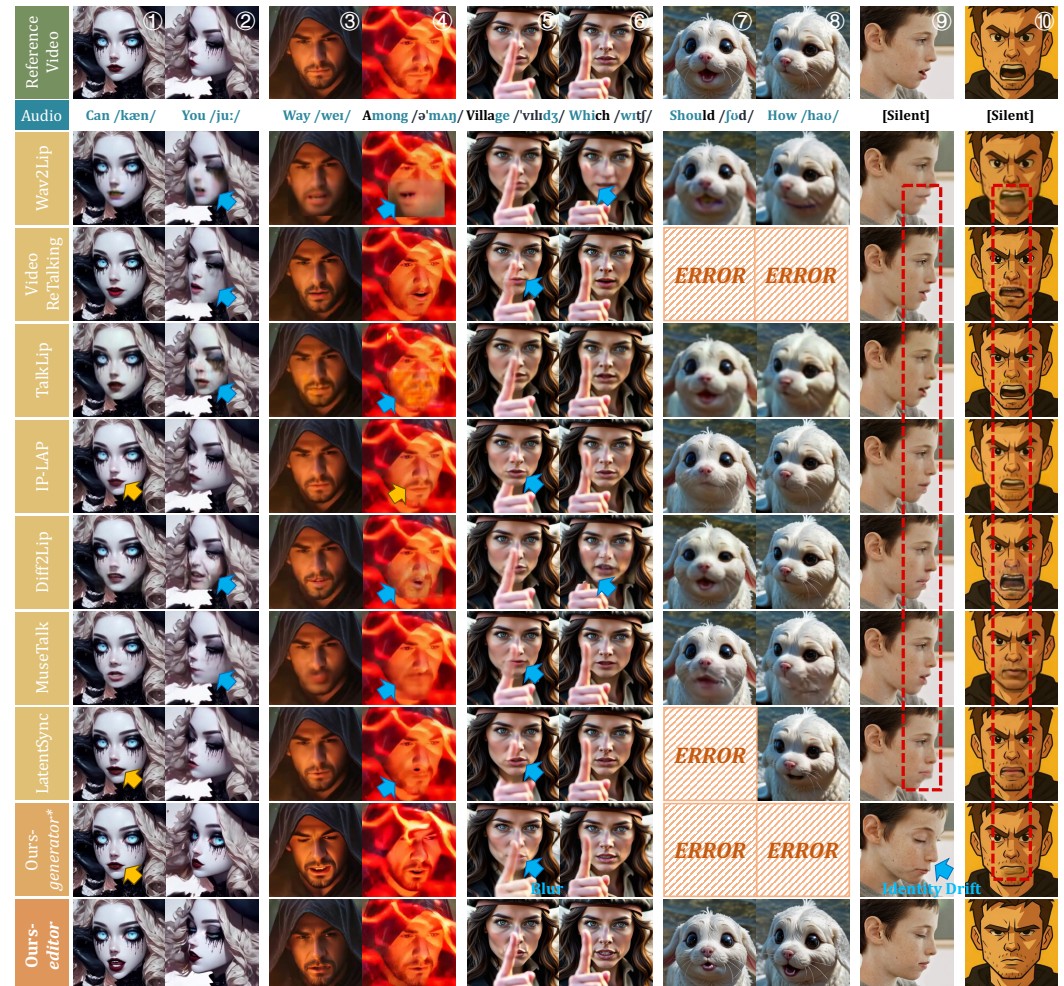

Figure 4: **Qualitative comparisons** across diverse scenarios. Lip-sync errors are marked with yellow, visual artifacts with blue, and lip leakage during silence with red. "ERROR" indicates runtime failure from missing 3DMM or landmarks despite best efforts. Our method exhibits robust performance with superior lip accuracy and identity consistency. Please **Qzoom in** for details.

Table 3: **User study results** of MOS with 95% confidence intervals.

| Method | Realism ↑ | Lip Sync ↑ | Identity ↑ | Overall ↑ |
|---|---|---|---|---|
| Wav2Lip | 2.56±0.11 | 2.80±0.13 | 3.07±0.14 | 2.35±0.10 |
| VideoReTalking | 3.00±0.09 | 3.09±0.11 | 3.58±0.09 | 3.22±0.11 |
| TalkLip | 2.59±0.13 | 2.08±0.11 | 3.06±0.11 | 2.73±0.11 |
| IP-LAP | 2.74±0.09 | 2.49±0.11 | 3.62±0.11 | 3.09±0.11 |
| Diff2Lip | 2.63±0.11 | 2.91±0.13 | 3.22±0.13 | 2.62±0.12 |
| MuseTalk | 2.45±0.10 | 2.35±0.11 | 2.98±0.14 | 2.49±0.11 |
| LatentSync | 2.91±0.11 | 2.81±0.12 | 3.62±0.11 | 3.16±0.13 |
| Ours-*generator*\* | 4.28±0.07 | 3.87±0.09 | 4.02±0.12 | 4.48±0.08 |
| **Ours-*editor*** | 4.40±0.06 | 4.50±0.06 | 4.40±0.07 | 4.66±0.05 |

Table 4: Our *editor* vs. constructed data.

| Method | FID ↓ | Sync-C ↑ | CSIM ↑ |
|---|---|---|---|
| Ours-*generator* (constructed data) | 7.00 | 7.88 | 0.905 |
| **Ours-*editor*** | 6.98 | 8.97 | 0.912 |

Table 5: **Ablation results** on HDTF dataset.

| Method | FID ↓ | Sync-C ↑ | LPIPS ↓ | CSIM ↑ |
|---|---|---|---|---|
| **Ours-*editor*** (full) | 7.03 | 8.56 | 0.014 | 0.883 |
| w/ channel concat | 6.89 | 7.49 | 0.014 | 0.873 |
| w/ uniform $t$ | 18.52 | 3.85 | 0.125 | 0.592 |
| w/o lip tuning | 7.00 | 7.68 | 0.013 | 0.875 |
| w/o texture tuning | 8.26 | 8.56 | 0.018 | 0.847 |

Quantitative results in Tab. 1 and 2 show that our *editor* sets a new state of the art. On HDTF, it achieves superior visual quality (FID –12.6%, FVD –25.0%), stronger lip sync (Sync-C +4.9%), and improved identity retention (CSIM +4.3%) over the best prior method. On the more challenging ContextDubBench, the advantages are even more pronounced: our model delivers better visual quality (NIQE 5.78 vs 6.11, BRISQUE 29.9 vs 39.2), higher lip–audio consistency (Sync-C +16.0%), and stronger identity preservation (CSIM +6.1%). Remarkably, it attains a success rate of 96.4%, exceeding the strongest baseline by over 24 points, while most prior methods remain around only

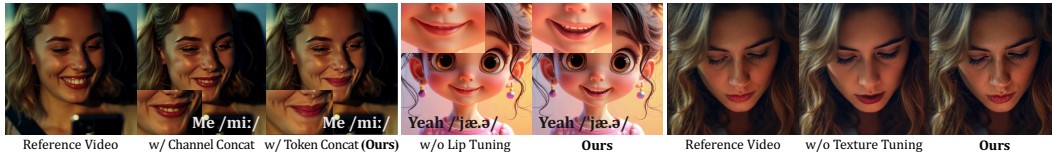

Figure 5: **Ablations on reference video conditioning and multi-phase learning.** Replacing our *frame-wise token-sequence concatenation* (abbr. concat) with *channel concat* for reference conditioning causes conflicts and thus lowers lip-sync accuracy. Removing the *lip* or *texture* phase degrades lip synchronization and detailed texture fidelity, respectively. Please 🔍**zoom in** for details.

60–70%. This large margin underscores the robustness and practical reliability of our approach in diverse and unconstrained scenarios, as the paired contextual inputs supply complete identity and spatiotemporal cues that allow the model to generalize beyond controlled settings.

Interestingly, our *generator\** already surpasses prior methods on HDTF, with clear gains in identity preservation (CLIPS +1.7%) and visual quality (FVD –26.8%). This highlights the strong generative capacity of the DiT backbone and its potential as a contextual synthesizer under tailored principles. More importantly, when trained on synthetic contextual pairs, our *editor* achieves further improvements (CSIM +3.3%, Sync-C +6.4%, and LPIPS –22.2%) while maintaining comparable FVD. These results demonstrate the effectiveness of our self-bootstrapping paradigm: the backbone not only generates paired data but also benefits from it, enabling stronger mask-free dubbing.

To further examine the self-bootstrapping effect, we evaluate the *generator* tailored for data construction. We sampled 20 synthetic pairs unseen during *editor* training, compared with the editor's outputs from the same inputs. As shown in Tab. 4, the *editor* consistently outperforms the constructed pairs in lip sync and visual quality. Notably, it even achieves stronger identity consistency than the training pairs. We attribute this to the fact that slight mismatches in synthetic companions, especially in fine-grained identity details, behave as speech-irrelevant noise that is suppressed during training. Meanwhile, the *editor* benefits far more from the rich, frame-aligned contextual signals than it is harmed by such noise, resulting in higher identity fidelity and stronger robustness.

## 4.2 QUALITATIVE EVALUATION

Fig. 4 shows qualitative comparisons, where our method consistently produces realistic, lip-synced results across challenging scenarios. Traditional baselines often yield inaccurate lip shapes (Col. 1), visual artifacts (Col. 2), weak robustness to occlusion (Col. 5), and side-view distortions with identity drift (Col. 2&9). Even our *generator\**, though using segmentation to handle occlusions, shows blur along mask boundaries and remains highly sensitive to mask accuracy. The rightmost column further reveals severe lip-shape leakage in all mask-based methods, where silent frames are corrupted by open-mouth artifacts. In contrast, our *editor* performs precise lip editing with faithful identity preservation and robustness to spatiotemporal dynamics (e.g., facial occlusions). Moreover, unlike some mask-based methods that rely on human-face priors, such as landmarks and 3D Morphable Models (3DMMs), to obtain facial masks and thus often fail on stylized or non-human characters (marked "ERROR"), our method uses contextual cues to localize speech-relevant regions without mask heuristics, yielding stable performance across character types and occlusions.

**User study.** We further conduct a user study with 30 participants on 24 dubbing videos from different methods, collecting Mean Opinion Scores (MOS). Each video is rated on a 5-point Likert scale for realism, lip sync, identity preservation, and overall quality. As shown in Tab. 3, our method holds clear margins over existing baselines across all aspects. Moreover, our *editor* surpasses *generator\**, particularly in identity consistency (4.40 vs. 4.02) and lip sync (4.50 vs. 3.87), validating the self-bootstrapping paradigm and yielding perceptually convincing, high-quality dubbing.

## 4.3 ABLATION STUDY

We conduct ablations on two key components: 1) reference video injection mechanism, and 2) timestep-adaptive multi-phase learning strategy, with results in Tab. 5 and visualization in Fig. 5.

For reference conditioning, replacing our frame-level token concatenation with channel concatenation causes a clear drop in lip sync (Sync-C –12.5%), which is also visible as lip-shape error in Fig. 5. Channel concatenation enforces rigid spatial fusion that conflicts with lip editing, while our token-based design uses self-attention to transfer identity without disturbing lips.

For training, replacing progressive multi-phase sampling with uniform timestep sampling, i.e., learning all noise levels at once, causes severe degradation and even divergence. Stage-wise comparisons further show that removing the lip phase reduces lip sync (–10.3%), with negligible gains in FID and LPIPS, while removing the texture phase weakens fidelity and identity (CSIM –4.1%). These results confirm that the three phases are complementary: high-noise pretraining secures global structure, mid-noise sharpens articulation, and low-noise restores textures and identity. Moreover, the progressive design eases contextual learning by allowing the model to address different information sequentially, rather than struggling with all aspects at once.

## 5 CONCLUSION

In this paper, we introduce a novel self-bootstrapping paradigm to address the core challenge in visual dubbing: the absence of paired real-world training data. We argue that instead of relying on masked inpainting, visual dubbing should be reframed as a well-conditioned video-to-video editing task. Built upon this paradigm, we present **X-Dub**, a context-rich dubbing framework, where a DiT model that first acts as a generator to create its own ideal training pairs with complete visual context, and then as an editor that learns from this curated data. This process is further refined by a timestep-adaptive multi-phase learning strategy that disentangles the learning of structure, lips, and texture, enhancing final output quality. Extensive experiments on standard datasets and our new challenging benchmark, ContextDubBench, demonstrate that our method achieves state-of-the-art results. X-Dub shows exceptional robustness in complex, in-the-wild scenarios, significantly outperforming prior works. We believe this work not only sets a new standard for visual dubbing but also offers a valuable insight for other conditional video editing tasks where paired data is scarce.

### ETHICS STATEMENT

This work presents a self-bootstrapping paradigm for visual dubbing, enabling more accurate and identity-preserving lip synchronization. While such technology can benefit applications in accessibility, education, and multilingual content production, it also raises ethical concerns. In particular, the ability to realistically alter speech and lip movements may facilitate misuse, including the generation of non-consensual content, impersonation, or misinformation. To mitigate these risks, we stress the importance of informed consent, respect for individual privacy, and transparent disclosure of synthetic media. Responsible deployment and adherence to ethical standards are crucial to ensure that advances in visual dubbing contribute positively to society.

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

# A  LLM Usage Statement.

We used large language models (LLMs) solely for linguistic assistance, such as grammar correction and style refinement. No part of the technical content, experimental design, analysis, or conclusions is generated by LLMs. The authors take full responsibility for the content of this paper.

# B  Details of Our DiT-Based Generator and Editor

## B.1  Preliminary of Flow-Matching-Based DiT Models

We adopt a pre-trained T2V DiT model as the backbone for both stages (Hong et al., 2022; Kong et al., 2024; Wan et al., 2025). It follows a latent diffusion paradigm with a 3D causal Variational Auto-Encoder (VAE) (Kingma & Welling, 2013) for video compression and a DiT (Peebles & Xie, 2023) for sequence modeling. Each DiT block interleaves 2D (spatial) self-attention, 3D (spatio-temporal) self-attention, text cross-attention, and feed-forward networks (FFN). Training follows standard flow matching (Esser et al., 2024; Lipman et al., 2022) with the forward process:

$$z_t = (1 - t)\, z_0 + t\, \epsilon, \quad \epsilon \sim \mathcal{N}(0, I), \tag{4}$$

and a v-prediction objective to predict $v = \epsilon - z_0$ conditioned on $c$:

$$\mathcal{L}_{\mathrm{FM}}(\theta) = \mathbb{E}_{z_0, \epsilon, t} \left[ \left\| v_\theta(z_t, t, c) - v \right\|_2^2 \right]. \tag{5}$$

## B.2  Adaptation of Text Cross-Attention Mechanism

To effectively adapt the pre-trained backbone—originally designed for text-to-video generation—for the visual dubbing task, we implement a specific strategy for handling text cross-attention. During training, we utilize Qwen2.5-VL Bai et al. (2025) to generate coarse captions for the target real videos, which serves to preserve the backbone's generative priors. However, to ensure the model primarily relies on the provided visual context and audio signals rather than textual descriptions, we apply a high dropout rate of 70% to these text conditions. Architecturally, the text embeddings interact exclusively with the noised target tokens via cross-attention, while the reference tokens remain unaffected. For inference, to maintain practical convenience without requiring user-provided captions, we employ an empty string as the positive prompt. Additionally, we leverage Classifier-Free Guidance (CFG) with standard negative prompts (e.g., "Blurry, deformed, low quality, distorted...") to suppress artifacts and ensure high-fidelity generation.

## B.3  Details of Our Mask-Based Generator

**Mask setting.** Previous mask-based dubbing methods typically employ either half-face rectangular masks from smoothly varying bounding boxes Prajwal et al. (2020); Cheng et al. (2022); Wang et al. (2023); Zhang et al. (2023) or fixed irregular-shaped masks on affine-transformed facial crops Guan et al. (2023); Li et al. (2024). However, the former's size variations often lead to lip motion information leakage, causing models to learn lip movements from visual occlusion changes rather than the conditional speech, resulting in shortcut learning. The latter constrains jaw position, disrupting pronounced mouth shapes such as wide-open expressions.

Instead, we utilize frame-wise estimated 3D Morphable Model (3DMM) (Retsinas et al., 2024) to obtain full-face masks. Specifically, we maintain each frame's pose, shape, and expression coefficients unchanged except for the jaw opening parameter, which is fixed at a maximum opening value of 0.4. We then project the facial mesh to generate masks. This approach minimizes mask size leakage caused by inter-frame lip variations while providing sufficient editable regions for unrestricted lip control. This strategy facilitates creating synthetic data with lip shapes distinct from the original video, aligning with our data construction principles.

**Audio conditioning.** Audio features are extracted using the Whisper (Radford et al., 2023) encoder and then injected via an audio cross-attention layer placed after text cross-attention. Since visual tokens and audio features have different temporal resolutions (1 video token frame corresponds to 8 audio-feature frames, i.e., 1:8), for each video frame, we select the corresponding audio feature

frames according to the timestamp, together with neighboring frames, forming a temporal window of size $n = 16$. This yields audio tokens $h_a \in \mathbb{R}^{(b \times f) \times n \times c}$, while video tokens are reshaped into $h_V \in \mathbb{R}^{(b \times f) \times (h' \times w') \times c}$, where $h' \times w'$ denotes the visual spatial size after patchification. Frame-wise cross-attention is then performed between the two modalities, where video tokens serve as queries and audio tokens as keys and values. Formally,

$$\text{Attn}(Q_V, K_A, V_A) = \text{softmax}\left(\frac{Q_V K_A^\top}{\sqrt{d}}\right) V_A, \quad Q_V = h_V W_Q^V, \ K_A = h_a W_K^A, \ V_A = h_a W_V^A. \tag{6}$$

**Reference conditioning.** The reference frame $I_{\text{ref}}$ is sampled from a different segment of the same video during training to prevent lip-shape leakage, while at inference from the target segment to provide visual cues under a similar head pose.

### B.4 DETAILS OF OUR CONTEXT-DRIVEN EDITOR

**3D Rotary Position Embedding (RoPE).** 3D RoPE is adopted in 3D self-attention of the DiT backbone to distinguish spatial-temporal positions, which we keep unchanged for target tokens. For reference tokens, inspired by Tan et al. (2024), we adapt RoPE to be temporally-aligned but spatially-shifted. Specifically, a reference token located at $(i, j, k)$, where $i$, $j$, and $k$ denote the height, width, and temporal indices, is mapped to $(i + h', j + w', k)$, with $(h', w', f')$ the spatial–temporal sizes after patchification. This design provides two benefits: **(1)** Temporal alignment enables frame-wise consistency preservation of dynamic attributes such as background and head poses; **(2)** Spatial shifting avoids direct overlap that could distort lip movements, and instead encourages the model to capture spatially misaligned yet correlated features like identity information.

## C DETAILS OF DATA CONSTRUCTION STRATEGIES

### C.1 SHORT-TERM SEGMENT PROCESSING

During *generator* inference with a single reference frame, we observe that denoising a long clip of 77 frames (matching the setting used by the backbone and the *editor*) in one pass causes noticeable texture and color drift in the tail frames relative to the first; the drift resets at the first frame of the next clip (see Fig. 6). Therefore, *under a single-reference regime, we conclude that single-pass denoising over long clips is detrimental to identity preservation.* We hypothesize two contributing factors: 1) the reference frame is anchored at the first position, so later frames become distant in the RoPE index space, amplifying identity drift; and 2) long clips naturally accumulate larger head motion and spatiotemporal changes, which a single reference frame cannot fully constrain.

To mitigate this, when constructing contextual pairs with the generator, we adopt short-segment training and inference: we generate clips of 25 frames and bridge adjacent clips with 5 motion frames, then concatenate them to form videos longer than 77 frames for supervising the *editor*, which is trained on 77-frame segments. This short-segment strategy enhances identity preservation, while any slight sacrifice in lip sync accuracy remains within our design guidelines, as shown in Tab. 6.

Table 6: Quantitative results between long-term and short-term processing.

| Method | Sync-C (Lip Sync) ↑ | CSIM (Identity Preservation) ↑ |
|---|---|---|
| Long-clip (77 frames) | 7.983 | 0.842 |
| Short-segment (25 frames, +5 overlap) | 7.841 | 0.867 |

### C.2 MASK PROCESSING WITH OCCLUSION HANDLING

To enhance the robustness of our generator against occlusions, namely, to maintain consistency with the original video's occlusion patterns and thereby facilitate the editor's ability to naturally inherit them, we introduce an occlusion-handling pipeline. First, a vision–language model (VLM) (Bai

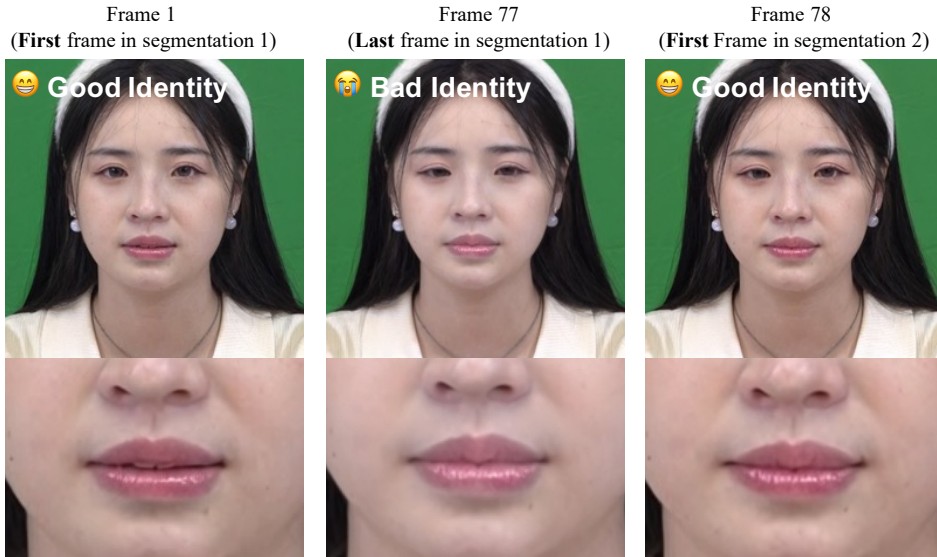

Figure 6: Example of intra-segment identity drift.

et al., 2025) is prompted per video with: *"Does any object occlude the person's face? If yes, output **only** a concise description of the object(s). If no, output nothing."* The returned object phrase(s) are then passed to SAM 2 (Ravi et al., 2024) to segment candidate occluders, yielding an occlusion mask $M_{\text{occ}}$. We apply a light manual screening step to remove severely erroneous segmentations.

Finally, we compose the occlusion-aware mask with the original inpainting mask. Let $M_{\text{face}}$ be the face mask (foreground 1, background 0), and $M_{\text{occ}}$ the occluder mask (1 on occluding objects). The visible-face mask is

$$M_{\text{vis}} = M_{\text{face}} \wedge \neg M_{\text{occ}},$$

and the inpainting mask (where *0* indicates regions to inpaint in our implementation) is

$$M_{\text{inp}} = \neg M_{\text{vis}} = \neg M_{\text{face}} \vee M_{\text{occ}},$$

where $\wedge$, $\vee$, and $\neg$ denote logical AND, OR, and NOT, respectively. As illustrated in Fig. 7, $M_{\text{inp}}$ excludes occluders while preserving non-occluded facial areas for inpainting.

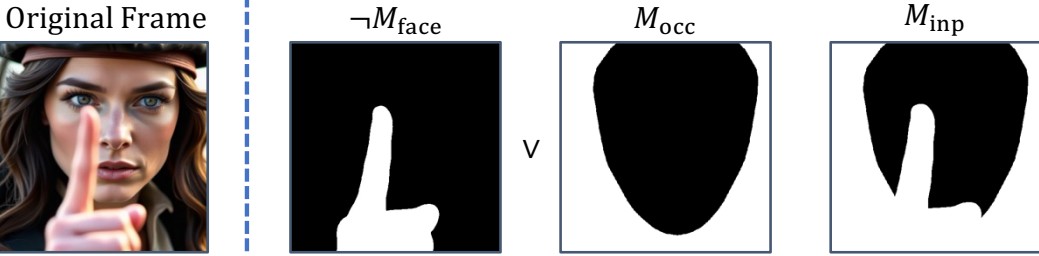

Figure 7: Example of mask processing with occlusion handling.

While our occlusion annotations can be incomplete or noisy, and using occlusion masks may introduce slight blur near mask boundaries and occasional lip degradation, as shown in the main text, the pipeline still supplies *paired and coherent* references that preserve the scene's occlusion patterns. This supervision encourages the final *editor* to model occlusion–face interactions in context, enabling robust handling of occlusions without labor-intensive manual intervention.

## C.3 LIGHTING AUGMENTATION

To enhance the robustness of our *editor* against challenging lighting conditions often encountered in real-world scenarios, we leverage an internal video relighting method to augment our paired training data. As illustrated in Fig. 8, we apply identical relighting effects to both the original target video $V$ and its synthetic companion $V'$. This synchronized processing ensures that the generated companion video continues to serve as a valid contextual reference, providing frame-aligned lighting cues that match the target. Our relighting strategy encompasses diverse configurations, including static illumination with varying chromaticities and intensities, as well as dynamic lighting effects. Specifically, to simulate complex environmental dynamics where the contextual input must provide time-varying illumination information, we introduce continuous variations in light source color, intensity, and direction across frames. To maintain high visual fidelity, we primarily apply this augmentation to high-quality video data captured in controlled, uniformly-lit laboratory environments. Furthermore, to prevent potential artifacts introduced by the relighting process itself from negatively impacting the *editor*'s training, we restrict this augmentation to a conservative fraction ($\sim$5%) of the total training dataset. This strategy effectively improves the model's generalization to in-the-wild lighting dynamics without compromising generation quality.

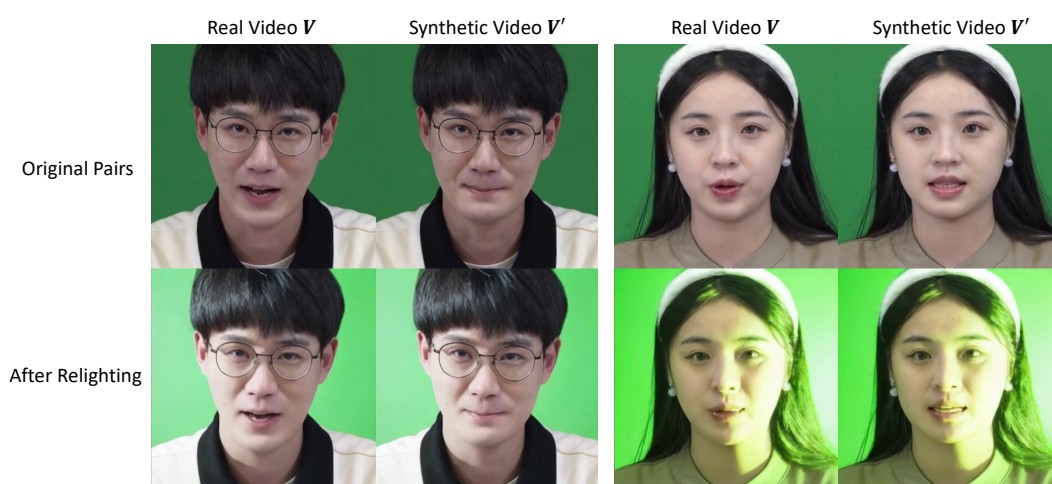

Figure 8: **Visualizing the lighting augmentation strategy.** We apply identical static and dynamic relighting effects (e.g., changing color, intensity, and direction) to both the generated contextual reference and the target video. This ensures the *editor* learns to utilize lighting cues from the context even under varying illumination conditions.

## C.4 POST-PROCESSING

Similar to Zhong et al. (2023), we use a Gaussian-smoothed face mask in post-processing to composite the generator's facial region back onto the original frames, mitigating minor background and boundary artifacts. Concretely, we blur the binary face mask $M_{\text{face}}$ with a Gaussian kernel to obtain $\tilde{M}_{\text{face}} \in [0, 1]$, and perform per-frame alpha blending:

$$V_{\text{post}} = \tilde{M}_{\text{face}} \odot V_{\text{gen}} + \left(1 - \tilde{M}_{\text{face}}\right) \odot V_{\text{orig}},$$

where $\odot$ denotes element-wise multiplication. This feathered composition keeps backgrounds consistent while preserving sharp facial edits, yielding training pairs with background-aligned context and helping the editor learn background-consistent editing behavior.

## C.5 QUALITY FILTERING

To maintain identity consistency while enforcing distinct lip shapes, we apply two complementary filters to each synthetic-original pair: **1) Identity similarity filter.** We use ArcFace (Deng et al., 2019) to compute cosine similarity between the synthetic and original videos. As a reference,

the mean within-speaker similarity across different real segments is 0.812, which is conservative given their differing head motions. Since our paired videos share identical head motion, we adopt a stricter threshold of 0.85 and discard pairs below this value to prevent identity drift. **2) Lip-shape distinction filter.** After aligning faces to a canonical template using the Umeyama algorithm following Deng et al. (2019), we measure the landmark distance over the mouth region between the original and synthetic videos. To ensure sufficient lip-shape variation, we reject pairs with a mouth-region landmark distance below 1.0.

To further safeguard visual quality and remove synthetic companions containing noticeable artifacts, we additionally assess each generated clip using a multimodal video-quality model Han et al. (2025). Each video is rated on six aspects including image fidelity, aesthetic appeal, temporal stability, motion smoothness, background consistency, and subject consistency, under a 5-point scoring scheme where 1 means very poor while 5 means excellent. We compute the average score across all six dimensions for each clip and retain only those with a mean score above 4.0, ensuring that only high-quality, artifact-free companion videos are included in the final training set.

### C.6  3D Talking Head Rendering Data

We leverage Unreal Engine to generate high-quality dubbing pairs. Initially, we acquire the 3D motion representation, which comprises ARKit-based facial expressions and 3D degree-of-freedom (3DOF) head poses. For each dataset entry containing speech audio and 3D motion representation $(A, M)$, we randomly select another entry $(A', M')$, and replace the speech-correlated coefficients in M with those from $M'$ to form $M_{dub}$. Both the original dataset entry and its corresponding dubbed version are rendered as follows:

$$V = \mathbf{R}(A, M, I),$$
$$V_{\text{dub}} = \mathbf{R}(A', M_{\text{dub}}, I), \tag{7}$$

where $\mathbf{R}$ denotes the Unreal Engine rendering pipeline (following Chen et al. (2025)) and $I$ represents the Unreal Engine MetaHuman avatar. To ensure data diversity, we create multiple avatars; however, it is important to note that the same avatar is used for each individual dubbing pair. Ultimately, we collect approximately 10 hours of 3D-rendered dubbing pairs in addition to the pairs generated by our DiT-based *generator*. These rendered pairs provide strictly aligned head motion, environment, and perfectly matched identity, which further enables the *editor* to focus on speech-related lip edits while preserving all other visual cues. Rendering examples are shown in Fig. 9.

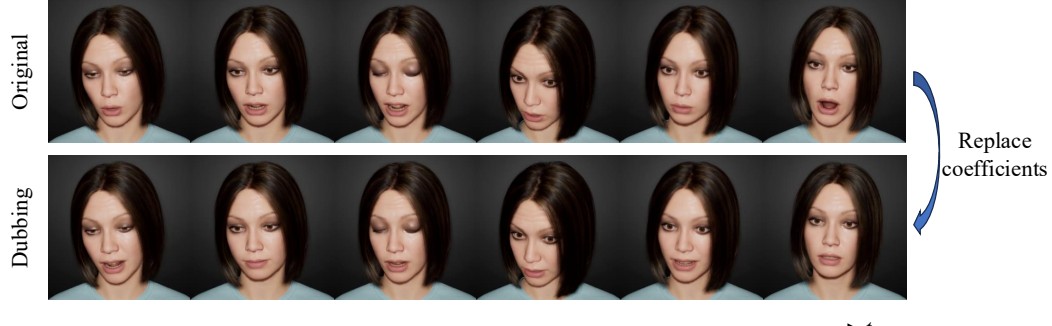

Figure 9: Example of aligned rendered video pairs.

## D  Details of Timestep-Adaptive Multi-Phase Learning

### D.1  Derivation of Eq. 3: Timestep-Constrained Single-Step Denoising

Given the forward diffusion process as in Eq. 4 and the v-prediction objective $v = \epsilon - z_0$, we derive the single-step denoising formula for pixel-level supervision during training, avoiding excessive computational overhead.

From Eq. 4, we can rearrange to obtain:

$$z_0 = \frac{z_t - t\,\epsilon}{1 - t}. \tag{8}$$

Since $\boldsymbol{v} = \boldsymbol{\epsilon} - \boldsymbol{z}_0$, we have $\boldsymbol{\epsilon} = \boldsymbol{v} + \boldsymbol{z}_0$. Substituting and solving for $\boldsymbol{z}_0$:

$$
\begin{aligned}
z_0 &= \frac{z_t - t(v + z_0)}{1 - t} = \frac{z_t - tv - tz_0}{1 - t} \\
(1 - t)z_0 &= z_t - tv - tz_0 \\
z_0 &= z_t - tv.
\end{aligned}
\tag{9}
$$

During inference, we use the predicted velocity $\hat{\boldsymbol{v}}$ instead of the true $\boldsymbol{v}$, yielding:

$$\hat{z}_0 = z_t - t\hat{v}. \tag{10}$$

Alternatively, we can express this as:

$$\hat{z}_0 = z_0 + t(v - \hat{v}), \tag{11}$$

which shows the reconstruction error depends on the velocity prediction error scaled by $t$.

However, when $t$ approaches 1, the velocity prediction error $(\boldsymbol{v} - \hat{\boldsymbol{v}})$ can be amplified, leading to poor reconstructions that result in inaccurate lip sync loss and identity loss computations. To address this, we introduce a timestep constraint:

$$\hat{x}_0 = \mathcal{D}(z_0 + (v - \hat{v}) \cdot \min\{t, t_{\text{thres}}\}), \tag{12}$$

where $\mathcal{D}$ denotes the VAE decoder. Importantly, this clipping is applied only in the denoising computation, not to the actual timestep $t$ used in the model's forward pass. The model still operates with the original $t$ value, enabling it to learn important structural and lip movement information in high- and mid-noise regions. We set $t_{\text{thres}} = 0.6$ in our experiments.

### D.2    SYNCNET SUPERVISION.

For lip-sync tuning, we adopt a SyncNet (Chung & Zisserman, 2016) comprising a visual encoder $S_V$ and an audio encoder $S_a$ to discriminate temporal alignment between video and audio clips. The lip-sync loss is defined as:

$$\mathcal{L}_{\text{sync}} = \text{CosSim}\big(S_V(\hat{x}_0^{[f:f+8]}),\ S_a(a^{[f:f+8]})\big). \tag{13}$$

This loss is combined with $\mathcal{L}_{\text{mFM}}$ defined in Eq. 1 in a weighted sum to train the lip-sync LoRA:

$$\mathcal{L}_{\text{total}} = (1 + w \cdot M + w_{\text{lip}} \cdot M_{\text{lip}}) \odot \mathcal{L}_{\text{FM}} + w_{\text{sync}} \cdot \mathcal{L}_{\text{sync}}. \tag{14}$$

### D.3    DETAILS OF PARAMETER CHOICES FOR MULTI-PHASE LEARNING.

Based on the well-established observation that diffusion models process information hierarchically Peng et al. (2025); Zhang et al. (2025); Meng et al. (2025), we first heuristically partition the noise schedule (timestep $t$) into three functional regions: high-noise for global structure, mid-noise for lip motion, and low-noise for detailed texture. This initial design is then finalized via the quantitative experiments.

**Determination of timestep shifting $\alpha$ for training.** To determine the optimal $\alpha$ values that allow the editor to efficiently learn decoupled information in each phase, and also to maximize the impact of the lip-sync and identity losses without degrading overall quality, we conduct an ablation study on the specific $\alpha$ settings. The experiments for the mid- and low-noise phases are conducted sequentially, each building upon the optimal parameter choice from the preceding stage.

The results are presented in Tab. 7. For the high-noise phase focused on global structure, we find that performance is relatively insensitive to $\alpha$ values above 3.0, with the model stably converging to high visual quality with minimal changes in FID and LPIPS. This is consistent with findings in Esser et al. (2024). We finally select $\alpha = 5.0$ to minimize overlap with the mid-noise phase.

Table 7: **Ablation study on the timestep shifting parameter $\alpha$ for each training phase on the HDTF dataset. Bold** indicates the best within a phase, while underline indicates the second best.

| Phase | $\alpha$ | Approximate Peak of $t$ | FID $\downarrow$ | LPIPS $\downarrow$ | Sync-C $\uparrow$ | Choices |
|---|---|---|---|---|---|---|
| **High-noise** | 5.0 | 0.921 | 8.25 | **0.017** | **7.68** | ✓ |
| | 4.0 | 0.899 | **8.24** | 0.018 | 7.64 | |
| | 3.0 | 0.861 | 8.31 | 0.020 | 7.65 | |
| **Mid-noise** | 3.0 | 0.861 | 10.52 | 0.021 | 8.47 | |
| | 2.0 | 0.777 | 8.39 | **0.017** | 8.50 | |
| | 1.5 | 0.684 | **8.26** | 0.018 | **8.56** | ✓ |
| | 0.8 | 0.392 | 8.31 | 0.018 | 7.21 | |
| **Low-noise** | 0.8 | 0.392 | 7.25 | 0.015 | 7.98 | |
| | 0.4 | 0.172 | **7.00** | 0.015 | 8.43 | |
| | 0.2 | 0.079 | 7.03 | **0.014** | **8.56** | ✓ |

Table 8: **Ablation study on the activation timestep ranges for LoRA experts during inference on the HDTF dataset. Bold** indicates the best, while underline indicates the second best.

| LoRA Expert | Timestep Range | FID $\downarrow$ | LPIPS $\downarrow$ | Sync-C $\uparrow$ | Choices |
|---|---|---|---|---|---|
| **Lip** | [0.0, 1.0] | 9.24 | 0.028 | 7.92 | |
| | [0.6, 1.0] | 8.52 | 0.020 | **8.61** | |
| | [0.4, 0.8] | **7.03** | **0.014** | 8.56 | ✓ |
| | [0.2, 0.6] | 7.26 | 0.016 | 8.03 | |
| **Texture** | [0.0, 1.0] | **6.95** | 0.015 | 6.74 | |
| | [0.1, 0.4] | 7.54 | 0.016 | 7.99 | |
| | [0.0, 0.3] | 7.03 | **0.014** | **8.56** | ✓ |

For the mid-noise tuning phase, an overly large $\alpha$ (e.g., 3.0) degrades overall visual quality, while a low value (e.g., 0.8) diminishes the effectiveness of the lip-sync loss. The model's performance is relatively stable within the intermediate range. We therefore select $\alpha = 1.5$ as the best balance between effective lip modification and preserving visual quality.

Finally, for the low-noise phase, a larger $\alpha$ (e.g., 0.8) disrupts the previously learned lip shapes. Smaller values are more stable, and based on our results, we select $\alpha = 0.2$. This choice allows the texture tuning to maximize its enhancement of visual quality while minimizing any negative impact on the already-learned lip motion.

**Determination of timestep intervals for LoRA expert activation for inference.** Similarly, we conduct an ablation study on the activation timestep ranges for the two trained LoRA experts during inference to determine the optimal phase boundaries. Note that this experiment uses the LoRA checkpoints trained with the optimal $\alpha$ values determined previously.

The results are shown in Tab. 8. First, for both experts, naively activating them across the entire denoising process ($t \in [0.0, 1.0]$) leads to a severe degradation in either visual quality or lip-sync, as this forces them to operate in timestep regions they rarely encountered during training.

For the lip LoRA expert trained in the mid-noise phase, activating it too early ($t \rightarrow 1$) conflicts with global visual quality and causes flickering artifacts around the mouth. Activating it too late ($t \rightarrow 0$) fails to sufficiently enhance lip sync. For the texture LoRA expert trained in the low-noise phase, activating it too early degrades the quality of the lip motion. Therefore, to balance these trade-offs, we select the non-overlapping ranges of $t \in [0.4, 0.8]$ for the lip LoRA and $t \in [0.0, 0.3]$ for the texture LoRA.

# E   OTHER IMPLEMENTATION DETAILS

We conduct experiments using a $\sim$1B-parameter T2V model on 32 A100 GPUs, with face-centered videos at $512 \times 512$ resolution and 25 fps. For the *generator*, we conduct extended training for $\sim$15k steps on 600 hours of internet audio-video data, sampling 25 frames with lr=1e-5 and batch size 256, which takes $\sim$1 day. Using the trained *generator*, we then synthesize the contextual video pairs, a

one-time data preparation step that takes ∼2 days. After inference and curation, we obtain 400 hours of video pairs, totaling 800 hours. For the *editor*, we begin with full-parameter training for ∼4k steps on 77-frame samples with lr=1e-5, batch size 256, and timestep shift $\alpha = 5.0$, followed by LoRA expert training for ∼1k steps each with lr=5e-6 and batch size 64; the entire training process for the *editor* takes ∼0.5 days. To reduce computational cost, we decode 4 tokens into 13-frame segments for pixel-level loss computation. Timestep shifts are set to $\alpha = 1.5$ for the lip expert and $\alpha = 0.2$ for the texture expert. Loss weights are set as $w = w_{\text{lip}} = 0.3$ for masks, and 0.05 for SyncNet, CLIP, and ArcFace loss.

## F  INFERENCE TIME AND COMPUTATIONAL COST

Inference with our *editor* requires approximately 30 GB of VRAM and fits comfortably on a single A100 GPU. With 50 denoising steps, the *editor* takes about 1 minute to process a 3-second, 25 fps video at $512 \times 512$ resolution.

To provide a comprehensive performance profile, we compare our *editor*'s inference time against some representative methods: a GAN-based dubbing model (Wav2Lip), a diffusion-based dubbing model (LatentSync), and a large-scale single-image animation model (MultiTalk). All diffusion-based methods are benchmarked with 50 denoising steps for a fair comparison, as shown in Tab. 9.

Table 9: Inference time comparison on a single A100 GPU. All diffusion models use 50 steps. The task is to process a 3-second, 25 fps video.

| Method | Wav2Lip | LatentSync | MultiTalk | **Ours-*editor*** |
|---|---|---|---|---|
| Model Type | GAN | Diffusion (UNet) | Diffusion (DiT) | Diffusion (DiT) |
| Parameters | ∼36M | ∼816M | ∼14B | ∼1.5B |
| Inference Time | ∼1s | ∼30s | ∼1800s (30 min) | ∼60s (1 min) |

The results in Tab. 9 show a clear quality-efficiency trade-off. As expected, our *editor* is slower than lightweight GAN-based methods like Wav2Lip, but its inference speed is comparable to other diffusion-based dubbing methods such as LatentSync. Crucially, our method achieves this comparable speed while delivering substantially better lip-sync accuracy and visual quality. Furthermore, when compared to large-scale animation models, our 1.5B parameter *editor* achieves an overall quality comparable to the 14B MultiTalk model, yet requires only a fraction of its inference time and parameter size. This demonstrates that a task-specific design for visual dubbing is more cost-effective than simply scaling up to a much larger, general-purpose video generation backbone.

Finally, our method can be significantly accelerated. Thanks to our context-rich editing formulation, the early, high-noise denoising steps primarily involve inheriting global structure from the input video. This allows us to safely reduce the number of denoising steps in this phase by approximately 10. Combined with lightweight acceleration techniques such as sequence parallelism and test-time caching (*e.g.*, TeaCache Liu et al. (2024)), we can further shorten the inference time to approximately 25 seconds for a 3-second clip, without noticeable quality degradation. This substantially mitigates practical deployment limitations.

## G  ABLATION ON PARADIGM VS. TRAINING STRATEGY

To clearly disentangle the contributions of our two core components—the self-bootstrapping paradigm (using constructed paired data) and the timestep-adaptive multi-phase learning strategy—we conduct a crucial ablation study. We compare four settings, varying the paradigm (inpainting vs. editing) and the training strategy (single-phase vs. multi-phase).

The results in Tab. 10 lead to two key findings. First, the primary performance gain stems from the paradigm shift enabled by the constructed paired data. Applying multi-phase learning to the inpainting-based *generator*\* (② vs. ①) yields negligible gains, and its performance remains far below that of our final *editor* (④). This demonstrates that the training strategy alone cannot overcome the fundamental limitations of the inpainting paradigm, which lacks frame-aligned visual context.

Table 10: Ablation study disentangling the contributions of the paradigm (inpainting vs. editing) and the training strategy (single-phase vs. multi-phase). Best results are in **bold**.

| | Method | Paradigm | Training Strategy | FID ↓ | LPIPS ↓ | Sync-C ↑ | Remarks |
|---|---|---|---|---|---|---|---|
| ① | generator* | Inpainting | Single-Phase (Uniform) | 7.87 | 0.018 | 8.05 | |
| ② | generator* | Inpainting | Multi-Phase | 7.92 | 0.018 | 8.19 | |
| ③ | editor | Editing | Single-Phase (Uniform) | 18.52 | 0.125 | 7.68 | Not converged. |
| ④ | **editor** | **Editing** | **Multi-Phase** | **7.03** | **0.014** | **8.56** | |

Second, the multi-phase learning strategy is an essential enabler for the *editor*, but not the *generator\**. The *generator\** converges well with uniform sampling (①) as its inpainting task is straightforward generation. The *editor*, in contrast, must balance the conflicting objectives of inheriting structure, editing lips, and preserving texture. A standard single-phase approach mixes these signals and causes training to collapse (③), whereas our multi-phase strategy disentangles them, enabling stable and effective training (④). In summary, the paradigm shift is the primary source of improvement, while the multi-phase learning strategy is a necessary mechanism that allows the *editor* to function reliably within this new paradigm.

## H  DETAILS OF USER STUDY

The user study involved 30 participants. Each participant received compensation of approximately 15 USD for completing a session that lasted 40–50 minutes, which aligns with the average hourly wage. For reference, Fig. 10 provides screenshots of the rating interface used in the study.

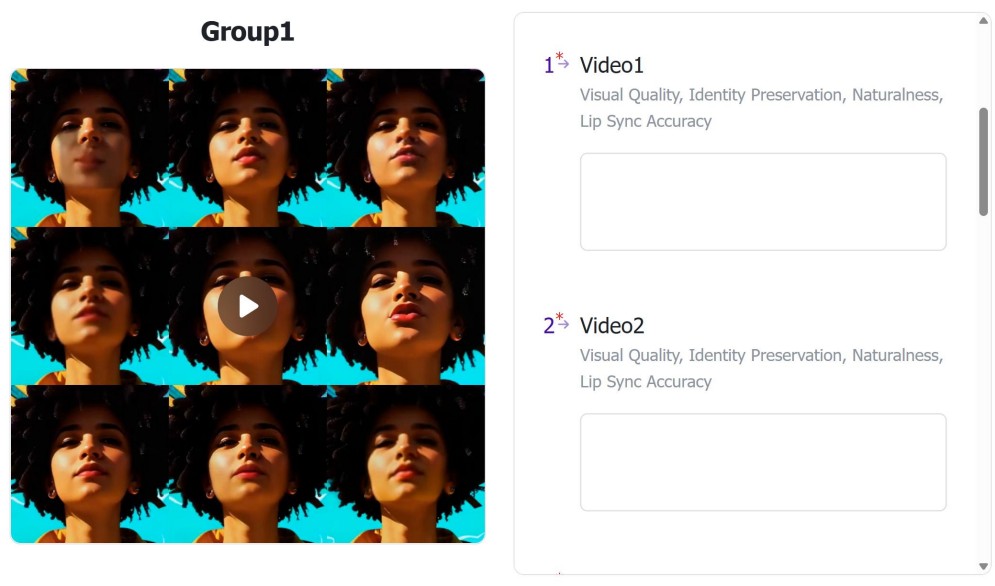

Figure 10: Screenshots of the rating interface of the user study.

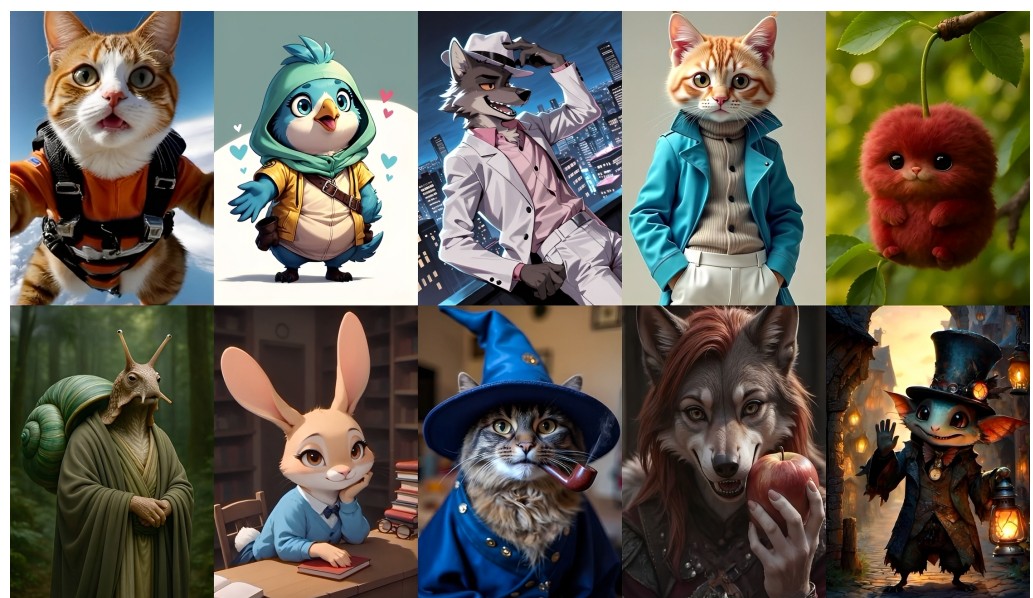

Figure 11: ContextDubBench benchmark Examples (I): Showcasing non-human characters with diverse morphological variations.

# I CONTEXTDUBBENCH

To thoroughly evaluate our framework, we construct ContextDubBench benchmark, a challenging benchmark comprising 440 video-audio pairs. The dataset is carefully designed with the following composition:

**Audio data.** The audio component includes both speech and singing. For speech, we randomly sampled 350 clips from Common Voice (Ardila et al., 2019), spanning six languages and dialects: 170 in English, 60 in Mandarin, 30 in Cantonese, 30 in Japanese, 30 in Russian, and 30 in French. For singing, we incorporated 60 English clips from NUS-48E (Duan et al., 2013) and 30 Mandarin clips from Opencpop (Wang et al., 2022). Each segment lasts between 7 and 14 seconds and captures a wide range of speaking rates, pitch levels, accents, and vocal styles, ensuring rich phonetic and linguistic diversity.

**Video data.** The video set combines real-world recordings and AI-generated content from publicly available sources with proper copyright clearance (e.g., Civitai, Mixkit, Pexels). It contains 291 clips of natural human subjects, 108 clips of stylized characters with distinct artistic features, and 41 clips of non-human or humanoid entities with durations ranging from 2 to 9 seconds. Representative samples are shown in Fig. 11, Fig. 12, and Fig. 13. Unlike conventional datasets, which are typically captured under controlled conditions, ContextDubBench is explicitly designed to reflect real-world challenges. The dataset incorporates dynamic lighting, partial occlusions, identity-preserving transformations, and substantial variations in pose and motion. By embedding these factors, ContextDubBench more faithfully captures the diversity and unpredictability of real-world scenarios, providing a rigorous testbed for evaluating lip-synchronization models. Illustrative examples are shown in Fig. 14.

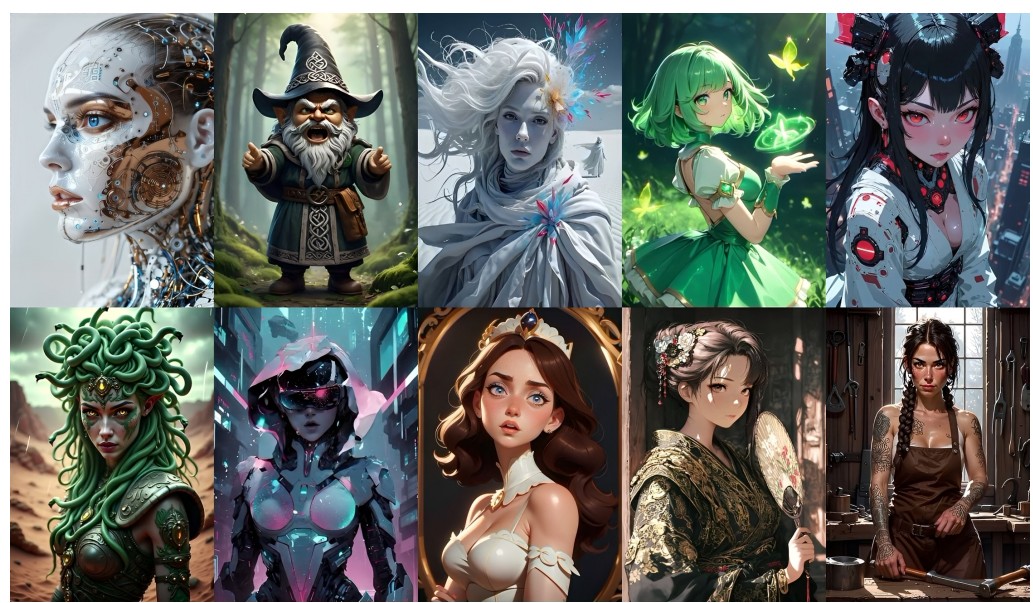

Figure 12: ContextDubBench benchmark Examples (II): Showcasing stylized characters with distinctive visual designs.

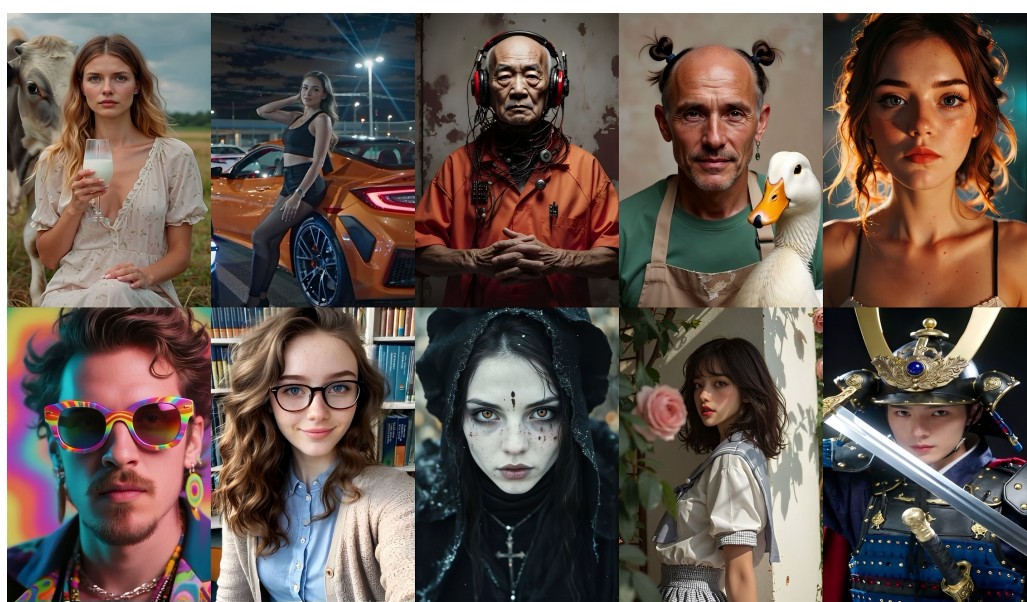

Figure 13: ContextDubBench benchmark Examples (III): Showcasing real-world human appearances in practical conditions.

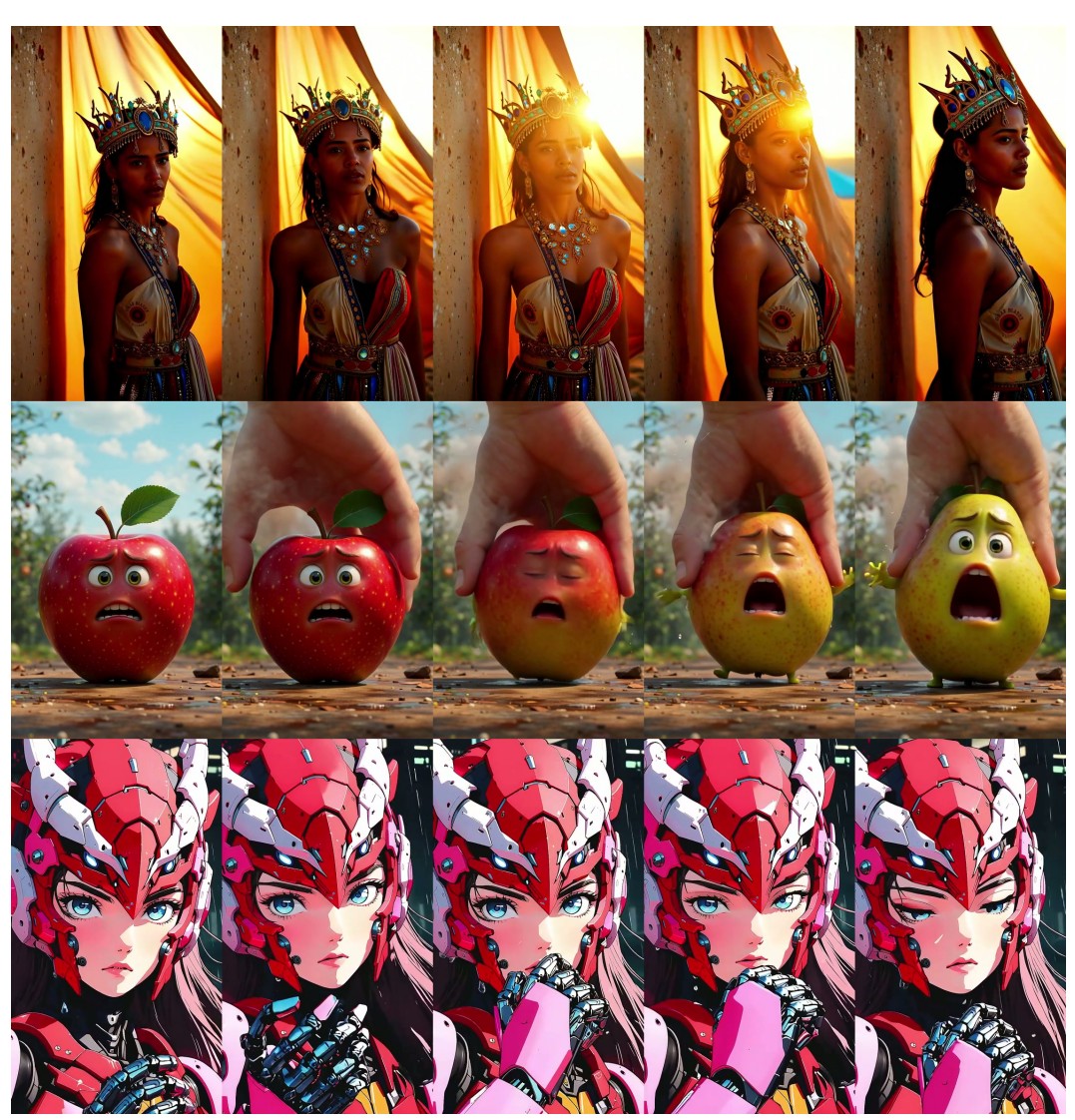

Figure 14: Samples from ContextDubBench benchmark showing lighting variations, identity-preserving changes, and occlusions, highlighting complex real-world scenarios.

