# OpenReview forum: "From Inpainting to Editing: A Self-Bootstrapping Paradigm for Context-Rich Visual Dubbing"
_ICLR.cc/2026/Conference — Submitted to ICLR 2026_

### Official Review · Reviewer_oymE · 2025-10-28

**Soundness:** 2
**Presentation:** 2
**Contribution:** 2
**Rating:** 4
**Confidence:** 3

**Summary:**

Based on the paper, a self-bootstrapping dubbing paradigm is proposed that leverages Diffusion Transformers (DiT) as both a generator of context-rich paired data and a video editor trained on them. This approach transforms dubbing from an under-specified inpainting task into a well-conditioned video-to-video editing problem. A timestep-adaptive multi-phase learning strategy is introduced to disentangle visual information learning across diffusion timesteps, facilitating more effective contextual learning and yielding enhanced lip-sync quality and visual coherence. Additionally, a new benchmark is proposed for evaluation.

**Strengths:**

The paper presents several notable strengths: it introduces a valuable benchmark for evaluation, features clear and well-structured writing, and demonstrates strong experimental design through comprehensive quantitative comparisons with solid evaluation metrics.

**Weaknesses:**

1. Limited Innovation: The core contribution appears to primarily reside in the benchmark development, while the video editing component relies mainly on some training strategies rather than substantial methodological breakthroughs.
2. Insufficient Supplementary Materials: The absence of supplementary video results restricts evaluation to single-frame qualitative analysis, which fails to adequately demonstrate the method's effectiveness, particularly as the presented results lack compelling visual evidence.

**Questions:**

1. What is the intrinsic relationship between the two different DiTs in relation to the task focused on in the paper?
2. Inference speed and resource requirements.

**Details Of Ethics Concerns:**

I don't have any concern.

---

> ### Author Response · Authors · 2025-11-21
> **Response to Reviewer oymE [Part 1/2]**
>
> Thanks for your constructive feedback and for recognizing the strengths in our benchmark design, writing clarity, and experimental setup.
> We appreciate the opportunity to clarify our methodological contributions and hope the following responses fully resolve your concerns.
>
> > **W1: About limited innovation and methodological contributions.**
>
> We appreciate your concern and believe it may stem from **a misunderstanding of our core methodological contributions** within the specific context of the **visual dubbing field**.
> To clarify, **our key contributions are:**
> - **A paradigm shift** reframing dubbing from ill-posed inpainting to well-conditioned video editing. This is enabled by a novel self-bootstrapping framework that systematically constructs the paired training data previously considered unattainable.
> - **A task-specific, timestep-adaptive training strategy** that ensures this paradigm's effectiveness by aligning dubbing goals with the hierarchical nature of diffusion models.
>
> The editing architecture itself serves merely as the vehicle for these contributions and is not claimed as the primary novelty. Similarly, our benchmark is specifically designed to validate these advances in complex dubbing scenarios. We detail why these core contributions constitute meaningful methodological advances below:
>
> **1) Visual dubbing is not a trivial editing task.**
> As highlighted in the **Introduction (Para. 1)**, this field is constrained by a fundamental, long-standing problem: the impossibility of capturing real-world, perfectly-aligned paired video data (the same person, under identical head pose and scene, speaking two different scripts). This has historically forced **all prior work into a compromised mask-inpainting paradigm**, which is fundamentally ill-posed due to incomplete and misaligned visual conditioning (i.e., context).
>
> **2)**
> Our framework is the first to systematically address this long-standing data bottleneck by synthesizing frame-aligned lip-altered companion videos using the powerful generative priors of a pretrained DiT video model.
> This conceptual advance is what **makes an editing-based approach to visual dubbing possible** and thus converts the problem into a well-conditioned task with full contextual information.
> Thus, our contribution is **enabling the shift to an editing perspective** (which is fundamentally new to visual dubbing, as proposed in the **Introduction, Para. 3**), rather than merely proposing a new editing architecture.
>
> **3)**
> Our timestep-adaptive learning is not a generic trick but **a necessary component for this paradigm**—a point emphasized in the **Introduction (Para. 5)** and empirically confirmed in **Table 5**.
> It leverages the diffusion hierarchy to disentangle the conflicting goals (inheriting structure, editing lips, and preserving texture) unique to this new editing formulation of visual dubbing.
>
> **In conclusion**, we believe that **reframing dubbing from inpainting to editing**, along with the synergistic training strategy that enables it, constitutes a substantial methodological contribution and provides a new perspective for the visual dubbing community.
> Beyond visual dubbing, these contributions may offer **broader insights**: our framework demonstrates **a viable strategy for editing tasks** where paired data is scarce, while our training approach exemplifies **how to practically leverage the diffusion hierarchy** to disentangle competing objectives.
>
> We have now **refined the phrasing** in the Abstract and Introduction of the revised paper to **more explicitly articulate** these unique challenges and our core methodological contributions.
>
> ---
> > **W2: Absence of supplementary video results.**
>
> We would like to respectfully clarify that a **demo video** was included in our initial submission via the **anonymous project page** ([x-dub-lab.github.io](https://x-dub-lab.github.io)) as referenced in the Abstract.
> For your convenience, we have now also included it directly in the [Supplementary Material](https://openreview.net/attachment?id=Q4zATLYoKi&name=supplementary_material).
> We warmly invite you to view it for a complete assessment, and would be happy to provide any additional explanation or video results upon your request.

---

> ### Author Response · Authors · 2025-11-21
> **Response to Reviewer oymE [Part 2/2]**
>
> ---
> > **Q1: About the intrinsic relationship between the two different DiTs.**
>
> Thank you for this insightful question. The two DiTs (i.e., the _generator_ and the _editor_) share the same DiT backbone structure  but play **distinct, complementary roles** that enable our paradigm shift:
>
> **1)** The _generator_ (**data synthesizer**) solves the data bottleneck. It operates on **incomplete context** (masked frames and misaligned references) to synthesize the "ideal" companion video $\boldsymbol{V}'$—creating pairs that share identical identity, pose, and background with the real video, differing only in lip motion.
>
> **2)** The _editor_ (**dubbing model**) learns the final dubbing task. It operates on **complete, aligned context** (the full $\boldsymbol{V}'$ during training, or the full input video at inference), leveraging strictly frame-aligned visual correspondence (e.g., lighting, texture, and occlusion details for every frame) to perform robust, high-fidelity editing.
>
> In summary, the _generator_ creates the necessary data conditions, while the _editor_ exploits them to solve the dubbing task from an editing perspective.
>
> ---
> > **Q2: Inference speed and resource requirements.**
>
> **1)** On a single A100 GPU and using a standard 50-step denoising schedule, our final dubbing model (the _editor_) takes **~1 minute** to synthesize a 3-second (25 fps) video clip, requiring around 30 GB of VRAM.
>
> **2)** Benefiting from our context-rich editing formulation, the model naturally inherits global structure in early steps. This allows us to skip ~10 high-noise steps with minimal quality loss. Combined with lightweight acceleration techniques (e.g., sequence parallelism and test-time caching such as TeaCache), the inference time can be **further reduced to ~25 seconds**.
>
> **3)** We provide a detailed comparison in Appendix F ("Inference Time and Computational Cost") of the revised version, demonstrating that our method achieves competitive runtime (e.g., LatentSync [R1] ~30 s) while delivering strong visual quality and lip-sync performance under the dubbing setting.
>
> ---
> **References**
>
> [R1]: C Li et al. LatentSync: Taming Audio-Conditioned Latent Diffusion Models for Lip Sync with SyncNet Supervision. arXiv 2025.
>
> ---
> We would be happy to provide any additional explanations or supplementary experiments upon request!

---

> > ### Comment · Reviewer_oymE · 2025-11-28
> > **Official Comment by Reviewer oymE**
> >
> > I appreciate the authors' detailed reply. Since the response has resolved most of my doubts, I have updated my rating to 6.
> >
> > ***
> > I am positive about the excellent visual results shown in the project page demos. After carefully reviewing the responses and re-examining the manuscript, I also acknowledge the work's contribution in establishing a new paradigm for visual dubbing by leveraging synthetic data to overcome the data bottleneck, as well as the value of the hierarchical multi-phase training strategy. These designs also offer valuable insights for other domains.
> > I have a few remaining minor points for further discussion:
> > 1.  While the inference speed is acceptable, what is the total time required for the training phase (including both stages) and data pair construction? This is crucial for understanding the method's scalability and deployment potential.
> > 2. Fig. 3 shows two cross-attention layers. Since the DiT backbone typically uses one layer for text interaction, and visual dubbing lacks text input, how is this handled? Clarifying this would help readers better understand the modification to the backbone.

---

> ### Author Response · Authors · 2025-11-29
> **Thanks for your comments!**
>
> Dear Reviewer oymE,
>
> We are delighted to hear that your concerns have been resolved, and we sincerely thank you for your **positive recommendation and decision to raise the rating**! Regarding your remaining minor points, we provide clarifications as follows.
>
> ---
> > **Q1: Total time for training and data construction.**
>
> As detailed in Appendix E ("Other Implementation Details"), our offline pipeline on 32 A100 GPUs requires **~1 day** for *generator* training, **~2 days** for data synthesis, and **~0.5 days** for *editor* training.
>
> While data construction is the most resource-intensive step, we emphasize that this is a **one-time offline process**. Once constructed, the dataset is stored and can be reused for all subsequent *editor* training and ablation experiments. Furthermore, this pipeline is **highly scalable**: more data can be incorporated simply by running the *generator* using the same offline procedure.
>
> ---
> > **Q2: About text cross-attention in the backbone.**
>
> Thank you for this insightful question regarding the architecture modification. We handle the text interaction as follows to adapt the pre-trained backbone for visual dubbing, details of which have been added to Appendix B.2 ("Adaptation of Text Cross-Attention Mechanism") in the revised version.
>
> **1) During training**, we use Qwen2.5-VL [R2] to generate coarse captions for the target real videos to preserve the backbone's generative priors, yet we apply a **high dropout rate (70%)** to these text conditions to force the model to rely primarily on visual context and audio. Note that only the noised target tokens interact with text embeddings via cross-attention, while reference tokens remain unaffected.
>
> **2) At inference**, to ensure practical convenience without requiring user-provided captions, we simply use an empty string as the positive prompt. We leverage Classifier-Free Guidance (CFG) [R3] with standard negative prompts (e.g., *"Blurry, deformed, low quality, distorted..."*) to ensure generation quality.
>
> ---
> **References**
>
> [R2]: S Bai et al. Qwen2.5-VL Technical Report. arXiv 2025.
>
> [R3]: J Ho et al. Classifier-Free Diffusion Guidance. NeurIPS Workshop 2021.
>
> ---
> We hope these clarifications fully address your remaining questions. Thank you again for your time and constructive feedback!

---

### Official Review · Reviewer_h2PZ · 2025-10-30

**Soundness:** 3
**Presentation:** 3
**Contribution:** 3
**Rating:** 6
**Confidence:** 3

**Summary:**

This paper proposes X-Dub, a self-bootstrapping diffusion framework that reframes visual dubbing from mask-based inpainting to context-rich video-to-video editing. A DiT-based generator first synthesizes paired videos with consistent context but varied lip motion, which are then used to train an editor for precise, audio-driven dubbing. The method achieves strong lip-sync accuracy, identity preservation, and robustness on both standard and challenging benchmarks.

**Strengths:**

1. The paper presents a novel paradigm that redefines visual dubbing as context-rich editing rather than inpainting, addressing the long-standing issue of incomplete contextual data.

2. The self-bootstrapping design elegantly generates synthetic paired data with high contextual consistency, enabling effective training without real-world paired supervision.

3. The proposed timestep-adaptive multi-phase strategy effectively disentangles global, lip, and texture information, leading to improved visual coherence and lip-sync precision.

4. The method demonstrates comprehensive and consistent performance gains.

**Weaknesses:**

1. The model’s dependence on self-generated training pairs may introduce domain bias and accumulate artifacts, limiting generalization to real-world data.

2. If the Generator produces mouth jitter or unnatural expressions, the Editor may learn to correct these artifacts rather than the true audio-driven dubbing mechanism.

3. Despite short-segment generation, long videos may still suffer from color or expression drift.

4. The Editor is trained on stable, noise-free, and pose-aligned pairs, which may reduce robustness under real-world conditions with occlusion, desynchronization, or compression noise.

5. The overall training pipeline is complicated, with high computational cost in multi-phase LoRA tuning and limited end-to-end optimization.

**Questions:**

1. How do the authors prevent artifacts or domain bias from self-generated training pairs from misleading the Editor?

2. Is the two-stage, multi-phase pipeline scalable, or could joint optimization simplify training?

---

> ### Author Response · Authors · 2025-11-21
> **Response to Reviewer h2PZ [Part 1/3]**
>
> We sincerely thank the reviewer for the thoughtful feedback and for acknowledging the novelty and strong performance of our method.
> We hope our responses below satisfactorily resolve all remaining concerns.
>
> ---
> >#### **W1-W4 & Q1: About Artifacts and Domain Bias from Self-Generated Data**
>
> We fully appreciate this concern. Mitigating the impact of potential domain bias and artifacts was indeed a core priority in our framework design.
> Below, we first clarify the general mechanism for preventing the model from learning synthetic artifacts (W1 & Q1), followed by specific responses for W2, W3, and W4.
>
> ---
> >**W1 & Q1: About preventing self-generated artifacts from misleading the _editor_ and limiting generalization to real-world data.**
>
> **1)**
> First and most fundamentally, our _editor_ is **always supervised by the real video ($\boldsymbol{V}$)**. The synthetic companion video ($\boldsymbol{V}'$) serves only as a conditional input (i.e., context), never as a learning target. This anchors the _editor_'s output distribution to real-world data and prevents the propagation of synthesis errors.
>
> **2)**
> We explicitly design the construction pipeline to **minimize artifacts** in $\boldsymbol{V}'$ and **maximize its visual alignment** with $\boldsymbol{V}$ via **principled construction heuristics** (e.g., short-segment generation to prevent drift), rigorous **quality filtering**, and realistic **augmentations** (details in Sec 3.1.2 and Appendix C). We also integrate a small set of perfectly-aligned 3D-rendered pairs to anchor precise lip editing.
>
> **3)**
> Any remaining minor artifacts in $\boldsymbol{V}'$ (e.g., slight jitter) are **stochastic and statistically uncorrelated with the audio**. They do not form a consistent, learnable pattern. The _editor_ naturally ignores this uninformative noise to focus on the robust, causal mapping from audio to lips. This is **empirically validated** in Tab. 4 of the main paper, where the _editor_'s output quality surpasses its own synthetic training inputs, providing compelling evidence that such stochastic synthetic noise does not mislead the editor's learning process.
>
> **4)**
> Ultimately, since perfect real-world paired data is physically unattainable, our approach represents the most effective solution. Extensive experiments confirm that the substantial **performance gains** from this paradigm shift—**enabled fundamentally by these synthetic pairs**—**far outweigh the negligible impact** of occasional synthetic noise.
>
> ---
> >**W2: "If the _generator_ produces mouth jitter or unnatural expressions, the _editor_ may learn to correct these artifacts rather than the true audio-driven dubbing mechanism."**
>
> This is an excellent point. As established in Point 3 of the response above, the only **systematic difference** between our constructed pairs ($\boldsymbol{V}'$ and $\boldsymbol{V}$) is the lip variation, which is strongly **correlated with the audio conditioning**.
> In contrast, any potential artifacts in $\boldsymbol{V}'$ like "mouth jitter" are **stochastic and audio-independent**.
> Therefore, correcting them does not form a consistent, learnable mapping.
> Consequently, the _editor_ is structurally driven to learn the robust, audio-correlated **dubbing behavior** rather than an unconditioned artifact-correction shortcut. **The successful learning of this mechanism is strongly evidenced by our superior lip-sync performance:** we achieve the highest Sync-C scores (measuring audio-visual synchronization) on both HDTF (Tab. 1) and ContextDubBench (Tab. 2), the best Lip Sync rating in the user study (Tab. 3), and qualitative video examples demonstrating high-precision, audio-driven lip editing.

---

> ### Author Response · Authors · 2025-11-21
> **Response to Reviewer h2PZ [Part 2/3]**
>
> ---
> >**W3 "Despite short-segment generation, long videos may still suffer from color or expression drift."**
>
> To clarify, the short-segment (25-frame) processing is applied **exclusively to the _generator_** during offline data creation to prevent drift in the synthetic context $\boldsymbol{V}'$. The final _editor_ operates directly on longer sequences (i.e., 77 frames) without this issue. We have further emphasized this in Sec. 3.1.2 (“Principled Pair Construction Strategies”) of the revised version to avoid similar confusion.
>
> **1)**
> The inpainting-based generator is indeed prone to color and identity drift when generating long sequences (e.g., 77 frames) in a single pass, due to the diminishing influence of the single static reference frame over time (as detailed in Appendix C.1).
> We therefore synthesize $\boldsymbol{V}'$ by stitching short 25-frame segments. By **frequently resetting the reference scope**, this strategy effectively prevents the **accumulation of generation errors**, ensuring $\boldsymbol{V}'$ provides highly consistent identity and color references throughout the video.
>
> **2)**
> The **_editor_** itself is trained and performs inference directly on full 77-frame clips. It is **naturally immune to drift** because our context-rich paradigm provides it with the complete, frame-aligned video as a constant source of stable identity and color information for every single frame. Theoretically, this allows us to dub videos of arbitrary length without instability; we now include a 1-minute dubbing example on our anonymous project page ([x-dub-lab.github.io](https://x-dub-lab.github.io)) to demonstrate this robustness, and we kindly invite you to take a look.
>
> ---
> > **W4 "The _editor_ is trained on stable, noise-free, and pose-aligned pairs, which may reduce robustness under real-world conditions with occlusion, desynchronization, or compression noise."**
>
> Thanks for your comment. We believe this concern arises from a potential misunderstanding of our task definition and data construction.
>
> **1)**
> First, training on **pose-aligned pairs** is not a limitation but a prerequisite for visual dubbing. The task definition explicitly requires modifying only the lips while strictly preserving head pose and background. Therefore, using aligned data is necessary to teach the model this precise editing behavior, rather than being a simplification.
>
> **2)**
> Our training pairs are actually **not noise-free or idealized**.
> The supervision target $\boldsymbol{V}$ is real-world footage containing natural blur, compression artifacts, and shadows.
> Our pipeline explicitly preserves these complexities in $\boldsymbol{V}'$ (e.g., using occlusion-aware masks and relighting augmentations) rather than removing them.
>
> Consequently, the _editor_ is trained on data rich with real-world challenges. In fact, **robustness is an inherent advantage of our context-rich paradigm**: by receiving the full, frame-aligned video—including real-world scene variations like occlusions—the _editor_ learns to preserve these factors naturally rather than hallucinate or inpaint them, leading to the strong and robust performance we demonstrate across diverse real-world scenarios.
>
> **Empirically**, our [demo video](https://x-dub-lab.github.io) showcases robustness against challenging occlusions (04:38-04:54), lighting conditions (04:57-05:05), and head pose changes (01:25-01:40). We have also added a new example to the [project page](https://x-dub-lab.github.io) demonstrating robustness to compression noise, and we warmly invite you to view it.

---

> ### Author Response · Authors · 2025-11-23
> **Response to Reviewer h2PZ [Part 3/3]**
>
> ---
> > **W5: About high computational cost in multi-phase LoRA tuning.**
>
> We thank the reviewer for this question. We would like to clarify that our multi-phase LoRA tuning is a strategy designed to **reduce training cost** and improve stability, rather than increasing it.
>
> Attempting to train our _editor_ in a single phase with uniform sampling is computationally prohibitive and unstable. Our experiments show this naive approach requires **~15k training steps** to reach an inferior result before the loss begins to diverge. In contrast, our multi-phase strategy requires only **~4k steps** for the initial full-parameter training, followed by just **~1k steps** for each lightweight LoRA. This totals only **~6k steps** to achieve a superior and more stable result, significantly reducing the overall computational budget. The **inference overhead is also minimal**, as it only additionally involves activating lightweight LoRAs during specific steps of the standard denoising process.
>
> ---
> > **W5 & Q2: About pipeline scalability and joint optimization.**
>
> Thanks for your very insightful comments!
>
> **1)**
> Our pipeline is **fully scalable and operationally simple** due to its **decoupled two-stage design**. The _generator_ is used only for a one-time, offline data creation process. The _editor_ is then trained end-to-end on this data separately.
> The framework scales naturally:
> - **More data** can be incorporated by running the _generator_ with the same offline procedure;
> - **Better _generator_** (or other backbone choices) can be plugged in to construct higher-quality pairs without changing the _editor_;
> - **The _editor_ itself can scale independently** in size or architecture while reusing the same stored paired data.
>
> **2)**
> **Joint optimization is unnecessary and likely undesirable** in our paradigm. The _generator_'s sole role is offline data creation; it has no direct influence on the _editor_'s learning loop. A joint approach would also be technically unstable, as the **diffusion-based _generator_** does not produce clean, fully-denoised frames during its own training. Our decoupled approach is a deliberate design choice that ensures a clean separation between data construction and the editing task.
>
> That being said, we **appreciate** the suggestion that **alternative formulations, including forms of joint optimization**, are a valuable direction for **future research** in visual dubbing. For our present work, however, the separation between data construction and editing has proven to be a stable, effective, and practical solution.
>
> ---
> Please don’t hesitate to let us know if there are any additional clarifications or experiments that we can offer!

---

### Official Review · Reviewer_Ji3y · 2025-10-31

**Soundness:** 3
**Presentation:** 3
**Contribution:** 3
**Rating:** 6
**Confidence:** 4

**Summary:**

This paper proposes X-Dub, a self-bootstrapping framework for audio-driven visual dubbing that shifts from traditional mask-based inpainting to context-rich video-to-video editing using Diffusion Transformers (DiTs). A DiT generator creates lip-altered companion videos to form paired training data with originals, enabling a DiT editor to focus on precise lip synchronization while preserving identity and handling challenges like occlusions and lighting variations. A timestep-adaptive multi-phase learning strategy aligns diffusion stages with global structure, lip shapes, and textures for enhanced quality. The work introduces ContextDubBench, a robust benchmark combining real and generated content. Experiments demonstrate state-of-the-art lip sync, identity fidelity, and robustness over baselines like Wav2Lip, Diff2Lip, and MuseTalk.

**Strengths:**

1. The problem is clearly defined — the paper nicely distinguishes visual dubbing from generic talking-head or animation generation, and provides a systematic analysis of why existing self-reconstruction based methods fail in this setting.
2. The paper is well written and easy to follow. The motivation, pipeline, and experiments are clearly presented, making it accessible even to non-specialists in video synthesis.
3. The visual results are impressive and effectively demonstrate the model’s advantages. The qualitative comparisons clearly highlight improvements in lip-sync accuracy and identity consistency.

**Weaknesses:**

1. The diffusion denoising process is hierarchical: early timesteps produce very coarse, blurry structure while fine details only emerge later. Lip-sync and identity losses, however, demand semantic-level precision. Applying those high-level losses too early can inject mismatched gradient signals, disturbing the coarse-stage optimization and causing blurriness, slow convergence, or instability.
2. The framework is two-stage (generator → editor), but it's unclear what exactly is contained in the promised “context.” It reads like the second stage simply fine-tunes or borrows features produced by the first stage — we need a clearer, concrete definition of what information the context carries and why that specific context is better than, say, stronger reference frames or alternative conditioning.
3. Using diffusion twice (generate contextual pair then edit) substantially increases compute. Compared to single-stage, end-to-end video generation/editing methods, the proposed pipeline may be heavier but the paper does not convincingly show a runtime or efficiency advantage — so the trade-off between quality and cost is unclear.

**Questions:**

1. Can you provide a more rigorous analysis of how lip-sync and identity losses interact with the diffusion timestep schedule? Right now the timestep choices look manual, please justify them theoretically or empirically, and show that these losses do not interfere destructively across timesteps.
2. Please provide concrete compute and latency numbers (FLOPs, GPU-hours, or wall-clock inference time) and clarify whether the method can meet real-time or near-real-time constraints. If not real-time, what are the practical deployment limits?

---

> ### Author Response · Authors · 2025-11-21
> **Response to Reviewer Ji3y [Part 1/2]**
>
> We sincerely thank the reviewer for your constructive suggestions and for recognizing our method's overall contribution.
> We hope our responses resolve your remaining concerns.
>
> ---
> > **W1 & Q1: Analysis of timestep choices.**
>
> Our choices are supported by both theoretical principles and empirical validation. Revised discussions have been added to Appendix D.3 (“Details of Parameter Choices for Multi-Phase Learning”).
>
> **1)**
> Based on the well-established principle that diffusion models process information hierarchically [R1,R2,R3], we **heuristically** partition timesteps $t$ into high (global structure), mid (lips), and low (texture) noise regions.
>
> **2)**
> For **training**, we adopt timestep shifting ($\alpha$) to concentrate $t$ sampling around specific peaks, thereby focusing the impact of semantic losses within targeted noise regions.
> Optimal $\alpha$ values are determined by the quantitative results on the HDTF dataset:
>
> | Phase | $\boldsymbol{\alpha}$ | Approx. Peak of $t$ | FID $\downarrow$| LPIPS $\downarrow$| Sync-C $\uparrow$| Remarks & Choice |
> |:-:|:-:|:-:|:-:|:-:|:-:|:-|
> | High-noise   | 5.0      | 0.921   | $\underline{8.25}$      | **0.017** | **7.68** |  Minimize phase overlap. ($\checkmark$) |
> |             | 4.0      | 0.899            | **8.24**   | $\underline{0.018}$        | 7.64    |          |
> |             | 3.0      | 0.861            | 8.31      | 0.020       | $\underline{7.65}$         |Insensitive for $\alpha > 3.0$.|
> | Mid-noise    | 3.0      | 0.861            | 10.52   | 0.021                      | 8.47                       |	Harms visual quality.|
> |             | 2.0      | 0.777            | 8.39                    | **0.017**                  | $\underline{8.50}$         |          |
> |             | 1.5      | 0.684            | **8.26**                | $\underline{0.018}$        | **8.56**                   | Best balance of sync and visual quality. ($\checkmark$) |
> |             | 0.8      | 0.392            | $\underline{8.31}$      | 0.018        | 7.21                       |Degrades lip sync.|
> | Low-noise    | 0.8      | 0.392            | 7.25                    | 0.015        | 7.98                       |	Disrupts learned lip sync.|
> |             | 0.4      | 0.172            | **7.00**                | $\underline{0.015}$        | $\underline{8.43}$         |          |
> |             | 0.2      | 0.079            | $\underline{7.03}$      | **0.014**                  | **8.56**                   | Best balance of texture and lip accuracy. ($\checkmark$) |
>
> *__Bold__ ($\underline{\text{underlined}}$): __best__ ($\underline{\text{second best}}$).
>
> **3)**
> At **inference**, the full DiT (high-noise phase) operates across all timesteps, while LoRA experts are selectively activated within their optimal timestep ranges, based on our analysis on the HDTF dataset:
>
> | LoRA Expert | Timestep Range | FID $\downarrow$ | LPIPS $\downarrow$ | Sync-C $\uparrow$ | Remarks & Choice |
> |:-:|:-:|:-:|:-:|:-:|:-|
> | Lip | [0.0, 1.0] (all) | 9.24 | 0.028 | 7.92 |Catastrophic collapse in visual quality.|
> |  | [0.6, 1.0] | 8.52 | 0.020 | **8.61** |Harms visual quality.|
> |  | [0.4, 0.8] | **7.03** | **0.014** | $\underline{8.56}$ | Optimal balance. ($\checkmark$) |
> |  | [0.2, 0.6] | $\underline{7.26}$ | $\underline{0.016}$ | 8.03 |Less effective lip sync.|
> | Texture | [0.0, 1.0] (all) | **6.95** | $\underline{0.015}$ | 6.74 |Destroys lip sync.|
> |  | [0.1, 0.4] | 7.54 | 0.016 | $\underline{7.99}$ |Destructive interference with lips.|
> |  | [0.0, 0.3] | $\underline{7.03}$ | **0.014** | **8.56** | Optimal balance. ($\checkmark$) |
>
> *__Bold__ ($\underline{\text{underlined}}$): __best__ ($\underline{\text{second best}}$).
>
> **Note:**
> Results confirm that our settings confine semantic losses and LoRA experts to their optimal functional regions, preventing destructive interference across timesteps.
>
> ---
> > **W1: Instability of applying semantic losses at early timesteps.**
>
> Thank you for this excellent point. Our design indeed prevents this through two specific mechanisms.
>
> **1)**
> The full DiT model is initially trained in the **high-noise phase using only the robust diffusion loss**. The lip-sync and identity losses are introduced only during LoRA tuning, where the training is concentrated in the more **stable mid- and low-noise regions**.
>
> **2)**
> We compute semantic losses on a **one-step denoised prediction** $\hat{\boldsymbol{x}}\_{0}$, not directly on noisy latents.
> To further stabilize $\hat{\boldsymbol{x}}\_{0}$, we use an **error-constrained denoising formula** (Eq. 3 in the main paper) that caps the error-scaling factor at a threshold $t_\text{thres}$.
> This ensures we obtain a coherent $\hat{\boldsymbol{x}}\_{0}$ clean enough for lip-sync supervision even at higher noise levels.
> In practice, $t\_\text{thres}=0.6$ prevents flickering artifacts seen with higher values, while ensuring effective lip supervision with stable gradients, without disrupting early-stage optimization.

---

> ### Author Response · Authors · 2025-11-21
> **Response to Reviewer Ji3y [Part 2/2]**
>
> ---
> > **W3 & Q2: About "using diffusion twice" and analysis of computational cost.**
>
> We appreciate your comments on computational cost.
>
> **1)**
> We would like to first clarify that **our framework is a two-stage training paradigm, not a two-pass inference system**. The _generator_ is used only once, offline, to create the training data. At inference time, **only the _editor_ model runs**, performing a single diffusion pass.
>
> **2)**
> The full details of our **training cost and resource usage** are provided in Appendix E ("Other Implementation Details").
> In brief: our offline **training** cost on 32 A100 GPUs involves ~1 day for the _generator_, ~2 days for data synthesis (a one-time cost), and ~0.5 days for the _editor_.
>
> **3)** At **inference** on a single A100 GPU (50 denoising steps), our _editor_ takes **~1 min** to process a 3-second, 25 fps video at 512x512 resolution, with ~192T FLOPs. As per your suggestion, we include a detailed inference speed comparison and analysis in Appendix F ("Inference Time and Computational Cost") in the revised version.
>
> | Method | Wav2Lip | LatentSync | MultiTalk | Ours-_editor_ |
> |:---|:---:|:---:|:---:|:---:|
> | Model Type | GAN | Diffusion (UNet) | Diffusion (DiT) | Diffusion (DiT) |
> | Parameters | ~36M | ~816M | ~14B | ~1.5B |
> | Inference Time (3s@25fps) | ~1s | ~30s | ~1800s (30 min) | ~60s (1 min) |
>
> **Analysis:**
> - Our method is slower than lightweight GAN-based methods, as expected, but is comparable in speed to other diffusion-based dubbing methods while offering a **superior quality-efficiency trade-off**.
> - It is also **far more efficient** than large-scale animation methods (MultiTalk), achieving comparable quality with a fraction of the parameters and runtime.
>
> **4)** Finally, our method can be efficiently accelerated. Thanks to our context-rich editing formulation, the early, high-noise denoising steps primarily involve inheriting global structure from the input video. This allows us to safely reduce the number of steps (by approx. 10 steps) in this phase. Combined with lightweight acceleration techniques (e.g., sequence parallelism and TeaCache), we can further **shorten the inference to ~25 seconds for a 3-second clip** without noticeable quality degradation. This substantially mitigates practical deployment limitations.
>
> ---
> > **W2: About a clearer definition and the superiority of the promised "context".**
>
> We appreciate the reviewer's valuable question, as the "context" is indeed at the core of our contribution. To address this, we have explicitly added a clearer definition of "context" to the Abstract and Introduction (Sec. 1) in our revised paper.
>
> **1）**
> In visual dubbing, we define "context" as all visual conditioning available to the model when synthesizing output frames:
> - **Context in mask-inpainting methods** consists of: a) __masked input video frames__ with the mouth area removed, yielding incomplete visual evidence; b) __one or a few static reference frames__ from other timestamps, which are misaligned in pose and scene with target frames.
> - **Context for the _editor_ in our paradigm** is the __full input video__ available during inference, which is complete, unmasked, and frame-aligned with the desired output.
>
> **2)**
> Our context provides the _editor_ with **all visual information from the input video**, excluding only the target lip motion. This includes **rich identity cues**, **complete scene interactions (e.g., lighting, occlusions, etc.)**, and **continuous spatiotemporal dynamics**, all perfectly aligned with the desired output frames.
>
> **3)**
> This is fundamentally **superior to any "stronger reference frame"** because our approach utilizes **all available frames as a complete, aligned, and dynamic conditioning signal**, leaving no missing regions or pose mismatches. By inheriting all of this visual evidence, the _editor_ only needs to modify the lip region, turning dubbing into a well-conditioned, local editing task. This is what enables high-fidelity and robust results even in challenging real-world scenarios with large motions, complex lighting, and occlusions.
> In contrast, inpainting methods must simultaneously guess appearance from misaligned references and hallucinate missing scene information.
>
> ---
> **References**
>
> [R1]: S Zhang et al. FlexiAct: Towards Flexible Action Control in Heterogeneous Scenarios. SIGGRAPH 2025.
>
> [R2]: Z Peng et al. Omnisync: Towards Universal Lip Synchronization via Diffusion Transformers. NeurIPS 2025.
>
> [R3]: C Liang et al. AlignHuman:​ Improving​ Motion​ and​ Fidelity​ via Timestep-Segment​ Preference​ Optimization​ for​ Audio-Driven​ Human​ Animation. arXiv 2025.
>
> ---
> We thank the reviewers again for the constructive feedback and are happy to provide any further clarifications or additional experiments if needed.

---

### Official Review · Reviewer_LNYW · 2025-11-01

**Soundness:** 3
**Presentation:** 3
**Contribution:** 3
**Rating:** 6
**Confidence:** 2

**Summary:**

This paper introduces a self-bootstrapping framework for talking-head generation, where a diffusion-based generator constructs synthetic contextual pairs to train an editor that performs high-quality dubbing without explicit masks. A timestep-adaptive multi-phase learning strategy is proposed to balance reconstruction, identity, and lip-sync objectives across different noise levels. The method shows strong results on standard benchmarks, demonstrating improved identity preservation and lip-sync accuracy.

**Strengths:**

- The paper presents strong quantitative results and competitive performance.
- It is clearly written and easy to follow.

**Weaknesses:**

1. The timestep-adaptive multi-phase learning strategy is central to the paper but not fully explained. It would be helpful to clarify how the phase ranges and α thresholds were chosen, and how different timesteps were selected for applying losses such as identity or lip-sync  loss. Additional quantitative or sensitivity analysis, or references supporting these design choices, would strengthen the methodology.
2. It is unclear how much of the improvement comes from the constructed paired data versus the timestep-adaptive multi-phase learning. The relative contribution of each factor is not explicitly analyzed, making it difficult to interpret the source of performance gains.

**Questions:**

1. Could the authors clarify how the phase boundaries and α thresholds were determined—for instance, whether they were selected through validation experiments or set heuristically?
2. Would applying the timestep-adaptive multi-phase learning to the generator* alone lead to comparable results? This could help clarify whether the performance gains mainly come from the timestep-adaptive learning or the constructed paired data.
3. It would be helpful if the authors could provide qualitative video examples on the HDTF benchmark.

---

> ### Author Response · Authors · 2025-11-21
> **Response to Reviewer LNYW [Part 1/2]**
>
> We sincerely thank the reviewer for your valuable feedback and for recognizing our method's effectiveness, clarity of writing, and overall contribution.
> We hope our responses fully resolve your remaining concerns.
>
> ---
>
> > **W1 & Q1: About clarification for parameter choices in the timestep-adaptive multi-phase learning strategy.**
>
> Thank you for this important question. Indeed, our $\alpha$ values and phase boundaries are determined by a two-step process: heuristic principles followed by empirical validation. The related discussion has been added to Appendix D.3 ("Details of Parameter Choices for Multi-Phase Learning") in the revised version.
>
> **1)** Based on the well-established observation that diffusion models process information hierarchically [R1,R2,R3], we first **heuristically** partition the noise schedule (timestep $t$) into **high-noise** (global structure), **mid-noise** (lip motion), and **low-noise** (texture refinement) regions. This provides the initial ranges for all parameters.
>
> **2)**
> For **training**, we follow [R4] to leverage timestep shifting ($\alpha$) to create "soft" partitions.
> This maintains $t$ sampling across the full [0, 1] range for stability, but concentrates the distribution around a peak (controlled by $\alpha$) to focus the impact of our phase-specific losses. The optimal $\alpha$ values were determined by the following analysis on the HDTF dataset:
>
> | Phase        | $\boldsymbol{\alpha}$ | Approx. Peak of $t$ | FID $\downarrow$     | LPIPS $\downarrow$          | Sync-C $\uparrow$         | Remarks & Choice |
> |:------------:|:--------:|:----------------:|:-----------------------:|:--------------------------:|:--------------------------:|:------|
> | High-noise   | 5.0      | 0.921            | $\underline{8.25}$      | **0.017**                  | **7.68**                   |  Minimize phase overlap. ($\checkmark$) |
> |             | 4.0      | 0.899            | **8.24**                | $\underline{0.018}$        | 7.64                       |          |
> |             | 3.0      | 0.861            | 8.31                    | 0.020                      | $\underline{7.65}$         |Insensitive for $\alpha > 3.0$.|
> | Mid-noise    | 3.0      | 0.861            | 10.52                   | 0.021                      | 8.47                       |	Harms visual quality.|
> |             | 2.0      | 0.777            | 8.39                    | **0.017**                  | $\underline{8.50}$         |          |
> |             | 1.5      | 0.684            | **8.26**                | $\underline{0.018}$        | **8.56**                   | Best balance of sync and visual quality. ($\checkmark$) |
> |             | 0.8      | 0.392            | $\underline{8.31}$      | 0.018        | 7.21                       |Degrades lip sync.|
> | Low-noise    | 0.8      | 0.392            | 7.25                    | 0.015        | 7.98                       |	Disrupts learned lip sync.|
> |             | 0.4      | 0.172            | **7.00**                | $\underline{0.015}$        | $\underline{8.43}$         |          |
> |             | 0.2      | 0.079            | $\underline{7.03}$      | **0.014**                  | **8.56**                   | Best balance of texture and lip accuracy. ($\checkmark$) |
>
> *__Bold__ ($\underline{\text{underlined}}$): __best__ ($\underline{\text{second best}}$).
>
> **3)**
> For **inference**, the full DiT (trained in the high-noise phase) functions across all timesteps. LoRA experts are activated within hard timestep boundaries to ensure they operate only in their optimal regions. The final ranges were determined by this analysis on the HDTF dataset:
>
> | LoRA Expert | Timestep Range | FID $\downarrow$ | LPIPS $\downarrow$ | Sync-C $\uparrow$ | Remarks & Choice |
> |:---:|:---:|:---:|:---:|:---:|:---|
> | Lip | [0.0, 1.0] (all) | 9.24 | 0.028 | 7.92 |Catastrophic collapse in visual quality.|
> |  | [0.6, 1.0] | 8.52 | 0.020 | **8.61** |Harms visual quality.|
> |  | [0.4, 0.8] | **7.03** | **0.014** | $\underline{8.56}$ | Optimal balance. ($\checkmark$) |
> |  | [0.2, 0.6] | $\underline{7.26}$ | $\underline{0.016}$ | 8.03 |Less effective lip sync.|
> | Texture | [0.0, 1.0] (all) | **6.95** | $\underline{0.015}$ | 6.74 |Destroys lip sync.|
> |  | [0.1, 0.4] | 7.54 | 0.016 | $\underline{7.99}$ |Destructive interference with lips.|
> |  | [0.0, 0.3] | $\underline{7.03}$ | **0.014** | **8.56** | Optimal balance. ($\checkmark$) |
>
> *__Bold__ ($\underline{\text{underlined}}$): __best__ ($\underline{\text{second best}}$).

---

> ### Author Response · Authors · 2025-11-21
> **Response to Reviewer LNYW [Part 2/2]**
>
> ---
>
> > **W2 & Q2: Relative contribution of constructed paired data vs. multi-phase Learning.**
>
> This is an insightful question regarding the relative contribution of our paradigm shift (using constructed paired data) versus the multi-phase learning strategy. As suggested, we now include this ablation study in Appendix G ("Ablation on Paradigm vs. Training Strategy") of the revised paper.
>
> | | Method | Paradigm | Training Strategy | FID $\downarrow$ | LPIPS $\downarrow$ | Sync-C $\uparrow$| Remarks |
> |:---:|:---:|:---:|:---:|:---:|:---:|:---:|:---:|
> | ① | _generator*_ | Inpainting | Single-Phase (Uniform) | 7.87 | 0.018 | 8.05 | |
> | ② | _generator*_ | Inpainting | Multi-Phase | 7.92 | 0.018 | 8.19 | |
> | ③ | _editor_ | Editing | Single-Phase (Uniform) | 18.52 | 0.125 | 7.68 | Not converged. |
> | ④ | _editor_ | Editing | Multi-Phase | **7.03** | **0.014**| **8.56** | |
>
> *__Bold__: __best__ .
>
> **Key Findings:**
> - The **primary performance gains stems from the constructed paired data**, which enables the paradigm shift from inpainting (_generator*_) to our context-rich editing (_editor_).
> Applying multi-phase learning to the _generator*_ gives no visual improvement and very minor lip-sync gains (② vs ①), remaining far below the performance of our final _editor_ (④), because the inpainting paradigm fundamentally lacks frame-aligned visual condition inputs (i.e., context).
>
> - **Multi-phase learning is an essential enabler for the _editor_, but not the _generator\*_.**
> The _generator*_ converges well with uniform sampling (①) as its inpainting task is straightforward generation.
> The _editor_, however, must balance conflicting objectives (inheritance, editing, and preservation). Uniform sampling mixes these signals and causes training to collapse (③), while our multi-phase strategy disentangles them, enabling stable and effective training (④).
>
> **In summary**, the paradigm shift enabled by the constructed paired data is the primary source of improvement. The multi-phase learning strategy is a necessary mechanism that allows the _editor_ to function reliably within this new paradigm, but it cannot by itself overcome the fundamental limitations of inpainting.
>
> ---
>
> > **Q3: About qualitative videos on the HDTF dataset.**
>
> Thank you for the reminder. Following your suggestion, we have now included qualitative video examples on the HDTF benchmark on our anonymous project page ([x-dub-lab.github.io](https://x-dub-lab.github.io)) and warmly invite you to view them for a more complete evaluation.
>
> ---
> **References**
>
> [R1]: S Zhang et al. FlexiAct: Towards Flexible Action Control in Heterogeneous Scenarios. SIGGRAPH 2025.
>
> [R2]: Z Peng et al. Omnisync: Towards Universal Lip Synchronization via Diffusion Transformers. NeurIPS 2025.
>
> [R3]: C Liang et al. AlignHuman:​ Improving​ Motion​ and​ Fidelity​ via Timestep-Segment​ Preference​ Optimization​ for​ Audio-Driven​ Human​ Animation. arXiv 2025.
>
> [R4]: P Esser et al. Scaling Rectified Flow Transformers for High-Resolution Image Synthesis. ICML 2024.
>
> ---
> Please feel free to let us know if there are any additional results or clarifications we can provide!

---

### Author Response · Authors · 2025-11-22
**General Response**

We sincerely thank all the reviewers for your constructive feedback and recognition of our work, especially for acknowledging the strengths of:

- **Novel paradigm with clear motivations** (Reviewer Ji3y, h2PZ),
- **Clearly defined problem and elegant self-bootstrapping solution** to the data bottleneck (Reviewer Ji3y, h2PZ),
- **Effective timestep-adaptive multi-phase strategy** for quality enhancement (Reviewer LNYW, Ji3y, h2PZ),
- **Comprehensive and consistent performance gains** with state-of-the-art results (Reviewer LNYW, Ji3y, h2PZ, oymE),
- **Impressive qualitative visual results** demonstrating the method's advantages (Reviewer LNYW, Ji3y),
- **Valuable benchmark** for evaluation in complex scenarios (Reviewer Ji3y, oymE),
- **Solid and extensive experimental design** to verify effectiveness (Reviewer Ji3y, oymE),
- **Well-written and easy to read** (Reviewer LNYW, Ji3y, h2PZ, oymE).


We have polished the paper, added the experiment results, and provided detailed clarifications in the revised version. Our manuscript is revised to include the following changes based on all the reviewers' insightful comments, which have helped us improve the paper quality significantly! Note that major updates in the main paper and appendix are highlighted in blue for better visualization.

- We have added the detailed parameter analysis for multi-phase learning in Appendix D.3 (Reviewer LNYW, Ji3y).
- We have included the ablation study on paradigm shift vs. training strategy in Appendix G (Reviewer LNYW).
- We have added the inference speed and cost analysis in Appendix F (Reviewer Ji3y, oymE).
- We have polished the Abstract and Section 1 to better articulate our core contributions and explicitly define "context" (Reviewer Ji3y, oymE).
- We have emphasized the short-segment processing details in Section 3.1.2 (Reviewer h2PZ).
- **[Dec. 3 Update]** We have clarified the adaptation of the text cross-attention mechanism in Appendix B.2 (Reviewer oymE).
- We have updated the anonymous [project page](https://x-dub-lab.github.io) with qualitative comparisons on the HDTF benchmark, long video generation, and robustness to compression noise (Reviewer LNYW, h2PZ).

Please don't hesitate to let us know if you have any additional comments on the manuscript.

---

### Author Response · Authors · 2025-11-27
**Welcome Further Discussion!**

Dear Reviewers,

We sincerely thank you again for your constructive feedback and recognition. Your insightful comments have been invaluable in helping us improve the quality of this paper.

We have provided point-by-point responses to address your specific concerns and accordingly polished the phrasing, added new experiment results, and provided necessary clarifications in the revised manuscript. We genuinely hope that these revisions effectively resolve your questions and further validate the contributions of our work.

**As the discussion period approaches its end**, we would like to take this opportunity to ensure that our responses are sufficient. **We remain fully available** to address any remaining concerns regarding methodology, implementation, or experiments. Please do not hesitate to let us know if further clarification is needed.

Best regards,

Paper 8838 Authors.

---

### Author Response · Authors · 2025-12-03
**Summary of Discussion & Positive Consensus**

Dear Area Chair,

We sincerely thank you and all reviewers for the dedicated time and constructive feedback. We are encouraged that the reviewers recognized the value of our work, specifically highlighting the **novel paradigm with clear motivations** [`Ji3y`, `h2PZ`; `oymE` (after discussion)], the **solid experimental design and extensive evaluation** [`Ji3y`, `oymE`], the **impressive visual results and competitive performance** [`LNYW`, `Ji3y`, `h2PZ`; `oymE` (after discussion)], and the **clarity of writing** [`All Reviewers`]. We have carefully addressed all concerns and achieved **positive feedback** during the discussion.

---
**Consensus: Scores Improved to All Positive**

Given the score reversion, we first clarify the **actual consensus** reached before the system reversion: the assessment evolved from an **initial {6, 6, 6, 4}** to **All Positive {6, 6, 6, 6}** (rating update to 6 explicitly stated in comments but blocked by the system).
Notably, Reviewer `oymE` (the sole initial negative rating of 4) acknowledged the contributions and visual quality of our work after the discussion, stating: _“Since the response has resolved most of my doubts, **I have updated my rating to 6**... I am positive about the visual results... acknowledge the work's contribution.”_

---
**Concerns and Resolutions**

As not all reviewers could participate in the final discussion, we summarize the key concerns and their resolutions below.

|Dimension|Key Concern|Our Response|Status|
|:---|:---|:---|:---|
|Contribution|Limited innovation without breakthroughs in the editing component (`oymE`)| Our core contributions are: **1) A paradigm shift** reframing dubbing from *ill-posed inpainting* to *well-conditioned editing* via self-generated data—providing rich context for robust, precise dubbing—rather than merely modifying the editing component. **2) A dedicated training strategy** leveraging diffusion hierarchy to disentangle conflicting editing objectives specific to dubbing, distinct from generic engineering tricks. |**✔ Resolved** — `oymE`: _"Acknowledge the work's contribution in establishing a new paradigm."_|
|Supplementary Material|Absence of video results (`oymE`)|The demo video was already **included in the initial submission** (linked in Abstract) but was initially overlooked.|**✔ Resolved** — `oymE`: _"Positive about the excellent visual results."_|
|Efficiency|Computational cost and inference speed (`Ji3y`, `oymE`)|Computational cost details (Appendix E) and inference speed analysis (Appendix F) demonstrate that our method offers **efficiency comparable to diffusion-based baselines** while delivering **superior quality**, with further potential for **acceleration** (optimizable to ~25s per 3s clip).| **✔ Resolved** — `oymE`: _"Inference speed is acceptable."_|
|Method Rationales|Choice of parameters for multi-phase learning (`LNYW`, `Ji3y`)|**Theoretical basis** (diffusion hierarchy) and supplemented **empirical ablation** (Appendix D.3) for the timestep-shifting parameter ($\alpha$) and phase boundaries justify our design choices, balancing visual quality and lip-sync accuracy.|**Addressed** (Detailed justification provided)|
|Synthetic Artifacts & Robustness|Potential synthetic artifacts and bias leading to error accumulation and limiting real-world robustness (`h2PZ`)| **Real-video supervision** and **principled data construction strategies**, alongside **rigorous filtering and augmentations**, minimize the impact of synthetic artifacts while preserving real-world challenges. Empirical results confirm that remaining stochastic artifacts are ignored, **causing no interference with the superior lip sync, visual quality, and in-the-wild robustness** enabled by our editing paradigm.| **Addressed** (Empirical and visual evidence provided)|
|Ablation & Attribution|Relative contribution of paradigm shift vs. training strategy (`LNYW`)|Quantitative ablation (Appendix G) identifies the **paradigm shift as the primary driver of improvement**, while the training strategy serves as a necessary mechanism to allow the editor to function reliably within this new paradigm.| **Addressed** (Quantitative analysis provided)|
|Concept Clarification|Definition of "context" and its superiority (`Ji3y`)| "Visual context" is explicitly defined as **all visual conditioning available** during synthesis. This distinguishes **fragmented inpainting inputs** from **complete, frame-aligned signals in our editing paradigm**, clarifying how the latter provides rich spatiotemporal information to transform dubbing into a well-conditioned task.|**Addressed** (Clarified in revised paper)|

Accordingly, we have revised the manuscript to incorporate additional experiments, clarify the methodology, and emphasize key contributions. We believe these refinements have significantly improved our work's quality. With **all reviewers now positively inclined**, we hope this summary helps facilitate your final decision.

Best regards,

Paper 8838 Authors

---

### Meta-Review · Area_Chair_5cJe · 2026-01-04

**Summary:**

This paper presents a self-bootstrapping framework for talking-head generation. A diffusion-based generator is used to create synthetic contextual pairs that train an editor capable of high-quality dubbing without relying on explicit masks. The method further introduces a timestep-adaptive, multi-phase training strategy designed to balance reconstruction fidelity, identity preservation, and lip-sync accuracy across different noise levels.

While the approach achieves strong empirical performance on standard benchmarks - particularly in terms of identity preservation and lip-sync accuracy - the methodological novelty from a machine learning perspective appears limited. As a result, the contribution feels incremental, and it is difficult to clearly identify the conceptual advances beyond quantitative improvements. In particular, the role and motivation of the second training phase are not sufficiently explained: it is unclear whether this phase serves as a refinement or fine-tuning stage, or why it is necessary at all. Although the authors addressed this point in the rebuttal, the explanation remains insufficiently clear.

**Reviewer Concerns:**

1. Insufficient motivation and limited methodological novelty - was not addressed by the authors
2. The inference procedure requires two sequential model runs, raising efficiency concerns - partially addressed by the authors
3. The reliance on self-generated training pairs may introduce domain bias and lead to the accumulation of artifacts -  was not addressed by the authors
4. Robustness for long-duration generations - addressed in the rebuttal

**Reviewer Scores:**

LNYW before: 6 -> after: 6

Ji3y before: 6 -> after: 6

h2PZ before: 6 -> after: 6

oymE before: 4 -> after: 6

---

### Decision · Program_Chairs · 2026-01-26

Reject